# GAS: Improving Discretization of Diffusion ODEs via Generalized Adversarial Solver

**Aleksandr Oganov**[*,1,†], **Ilya Bykov**[*,1], **Eva Neudachina**[*,1], **Mishan Aliev**[1],
**Alexander Tolmachev**[4], **Alexander Sidorov**, **Aleksandr Zuev**,
**Andrey Okhotin**[1,2], **Denis Rakitin**[1], **Aibek Alanov**[1,3]

[1]HSE University, Russia
[2]Lomonosov Moscow State University, Russia
[3]FusionBrain Lab, AXXX, Russia
[4]Moscow Independent Research Institute of Artificial Intelligence, Russia

## Abstract

While diffusion models achieve state-of-the-art generation quality, they still suffer from computationally expensive sampling. Recent works address this issue with gradient-based optimization methods that distill a few-step ODE diffusion solver from the full sampling process, reducing the number of function evaluations from dozens to just a few. However, these approaches often rely on intricate training techniques and do not explicitly focus on preserving fine-grained details. In this paper, we introduce the *Generalized Solver*: a simple parameterization of the ODE sampler that does not require additional training tricks and improves quality over existing approaches. We further combine the original distillation loss with adversarial training, which mitigates artifacts and enhances detail fidelity. We call the resulting method the *Generalized Adversarial Solver* and demonstrate its superior performance compared to existing solver training methods under similar resource constraints. Code is available at `https://github.com/3145tttt/GAS`.

## 1 Introduction

Diffusion models (Sohl-Dickstein et al., 2015; Ho et al., 2020; Song et al., 2020b) offer state-of-the-art generation quality in diverse vision problems, including unconditional and conditional (Dhariwal & Nichol, 2021; Ho & Salimans, 2022) generation, text-to-image (Nichol et al., 2021; Ramesh et al., 2022; Saharia et al., 2022; Rombach et al., 2022; Esser et al., 2024), text-to-video (Blattmann et al., 2023; Brooks et al., 2024; Zheng et al., 2024; Chen et al., 2024b) and even text-to-3D (Poole et al., 2022; Wang et al., 2023) generation. One of the reasons for their success consists in satisfying both high sample quality (Dhariwal & Nichol, 2021; Karras et al., 2022) and mode coverage from the generative trilemma (Xiao et al., 2021). In theory, this allows diffusion models to produce desirable samples from the target distribution given unlimited computation time.

Besides, many improvements were made to satisfy the third requirement on generation speed. One way to tackle high inference time is to train a new model that utilizes the pre-trained diffusion and requires fewer inference steps. This may be achieved by straightening the generation trajectories (Liu et al., 2022b; 2023; Wang et al., 2024) or by directly performing diffusion distillation (Salimans & Ho, 2022; Song et al., 2023; Sauer et al., 2023; Yin et al., 2023) into a few-step student. These training-based methods are capable of fast generation with superior quality on large-scale scenarios. Their training procedures, however, are computation and memory-heavy and may be infeasible for users with resource constraints on cutting-edge problems, such as video generation.

Due to the mentioned resource requirements, the lightweight approach of directly accelerating generation is preferable most of the time. Such inference-time methods as designing specific solvers (Song et al., 2020a; Lu et al., 2022a; Zhang & Chen, 2022), caching intermediate steps (Ma et al., 2024; Wimbauer et al., 2024), or performing quantization (Gu et al., 2022; Badri & Shaji, 2023), push the boundaries of the pre-trained model by utilizing its knowledge as much as possible given

---

[*]Equal contribution. [†]Correspondence: `3145tttt@gmail.com`

Figure 1: Illustration of the **Generalized Adversarial Solver** image generation in comparison with the training-free UniPC (Zhao et al., 2024) solver with equal number of function evaluations (NFE). Our method shows superior results that are almost identical to teacher images in terms of generation quality.

a fixed computational budget. Among them, specifically designed solvers are mostly theoretically sound and are capable of producing high-quality samples similar to the full-inference model. However, they require significant hyperparameter search (Zhou et al., 2024b; Zhao et al., 2024) for each model and may be suboptimal depending on the particular setting.

A natural improvement of the idea consists of training (hyper-)parameters of the inference-time "student" sampler to match the full-inference "teacher" model. The approach is free-form and allows for optimizing timestep schedule (Sabour et al., 2024; Tong et al., 2024) as well as the sampler coefficients (Kim et al., 2024; Frankel et al., 2025) for each prediction step. Currently existing methods for training the sampler succeed in improving test-time efficiency of the model compared to the standard solvers. At the same time, they do not realize the full potential of the paradigm and tend to have inefficiencies that lead to nuanced and complicated training schemes. Among these are the unstable loss scale (Sabour et al., 2024), limited parameter space (Tong et al., 2024) and disentanglement of the parameter subsets (Frankel et al., 2025), which we find to be harmful for training. Besides, straightforward sampler distillation into a student with limited parameters may be ineffective for preserving the fine-grained details and may interfere with the generation quality.

In this paper, we aim to tackle the aforementioned issues by introducing a simple yet effective sampler parameterization and modifying the distillation loss. Specifically, we construct a sampler that performs each sampling step by calculating a weighted sum of the current velocity direction with all of the points and directions from previous steps. We propose to utilize a pre-defined solver as a time-dependent guidance and learn correction to its theoretically derived weights to facilitate and accelerate training. On top of that, we endow the sampler distillation with the adversarial loss (Goodfellow et al., 2014) to further boost the sampler quality. Most importantly, we

1. Introduce a novel sampler parameterization that we call the *Generalized Solver* and demonstrate its significant impact on training acceleration;

2. Combine it with the adversarial training and validate its positive impact on the fine-grained generation details;

3. Show that the resulting *Generalized Adversarial Solver* achieves superior results compared to the existing methods of solver/timestep training on several pixel-space and latent-space data sets.

## 2 BACKGROUND

### 2.1 DIFFUSION MODELS

Diffusion models (Sohl-Dickstein et al., 2015; Ho et al., 2020; Song et al., 2020b) simulate the data distribution by defining the forward process of gradual data noising and constructing its time reversal. The *forward* process is commonly defined by a sequence $\{p_{t|0}\}_{t \in [0,T]}$ of transition probabilities $p_{t|0}(\mathbf{x}_t|\mathbf{x}_0) = \mathcal{N}\left(\mathbf{x}_t \mid \alpha_t \mathbf{x}_0, \sigma_t^2 \mathbf{I}\right)$. It perturbs the initial data distribution $p_{\text{data}}(\mathbf{x}_0) = p_0(\mathbf{x}_0)$ by destroying part of its signal and replacing it with the independent Gaussian noise. Here, $\alpha_t$ and $\sigma_t^2$ are positive differentiable functions that define the corresponding *noise schedule*. Typically, their choice ensures that the sequence of the corresponding marginal distributions $p_t(\mathbf{x}_t)$ converges to a simple and tractable prior distribution $p_T(\mathbf{x}_T)$ (e.g. standard normal). For each noise schedule one

can construct the equivalent Probability Flow ODE (PF-ODE) (Song et al., 2020b)

$$d\mathbf{x}_t = \left[ f(t)\mathbf{x}_t - \frac{1}{2}g^2(t)\nabla_{\mathbf{x}_t} \log p_t(\mathbf{x}_t) \right] dt, \tag{1}$$

where setting

$$f(t) = \frac{d \log \alpha_t}{dt}, \quad g^2(t) = d\sigma_t^2 - 2\frac{d \log \alpha_t}{dt}\sigma_t^2 \tag{2}$$

and sampling the endpoint $\mathbf{x}_T$ from the prior distribution $p_T$ ensures (Lu et al., 2022a) that $\mathbf{x}_t \sim p_t$ for all timesteps. Essentially, ODE formulation allows one to obtain a *backward* process of data generation by reversing the velocity of the particle given access to the *score function* $\nabla_{\mathbf{x}_t} \log p_t(\mathbf{x}_t)$ of the perturbed data distribution. In practice, diffusion models approximate the score function by optimizing the Denoising Score Matching (Vincent, 2011) objective

$$\min_\theta \int_0^T \mathbb{E}_{p_{0,t}(\mathbf{x}_0,\mathbf{x}_t)} \| s_\theta(\mathbf{x}_t, t) - \nabla_{\mathbf{x}_t} \log p_{t|0}(\mathbf{x}_t|\mathbf{x}_0) \|^2 dt, \tag{3}$$

where the score functions $\nabla_{\mathbf{x}_t} \log p_{t|0}(\mathbf{x}_t|\mathbf{x}_0)$ of the conditional Gaussian distributions are tractable and equal to $-(\mathbf{x}_t - \alpha_t \mathbf{x}_0)/\sigma_t^2$. Besides the score networks, one can directly approximate the ODE velocity function by setting $\boldsymbol{v}_\theta(\mathbf{x}_t, t) = f(t)\mathbf{x}_t - (1/2)g^2(t)s_\theta(\mathbf{x}_t, t)$.

## 2.2 ODE SOLVERS

Sampling from a diffusion model amounts to numerically approximating the solution of the corresponding PF-ODE (Eq. 1). Standard numerical methods for solving a general-form ODE $d\mathbf{x}_t = \boldsymbol{v}(\mathbf{x}_t, t)dt$ are mainly based on approximating the direction $\mathbf{x}_{t+h} - \mathbf{x}_t$ via Taylor expansion.

The first-order Euler scheme makes a step $h \cdot \boldsymbol{v}(\mathbf{x}_t, t)$, which is simple, yet has a large discretization error. Its higher-order modifications generally approximate the derivatives with finite differences. This correction allows Runge-Kutta methods to produce high-quality results (Lu et al., 2022a; Zhang & Chen, 2022; Karras et al., 2022). However, these methods require mid-point evaluations, which harms performance in low-NFE regimes (see e.g. (Zhang & Chen, 2022, Table 2)). In contrast, Linear Multistep solvers (Liu et al., 2022a; Zhang & Chen, 2022) use only previously calculated points and directions for the same approximation, thus remain useful in this setting.

Recently designed solvers such as DDIM (Song et al., 2020a), DPM-Solver(++) (Lu et al., 2022a;b), DEIS (Zhang & Chen, 2022), and UniPC (Zhao et al., 2024), exploit the semi-linear nature of the PF-ODE (Hochbruck & Ostermann, 2010). They approximate the integral in the "variation of constants" formula

$$\mathbf{x}_t = \frac{\alpha_t}{\alpha_u}\mathbf{x}_u - \int_u^t \frac{\alpha_t}{\alpha_\tau} \cdot \frac{g^2(\tau)}{2} s(\mathbf{x}_\tau, \tau)d\tau, \tag{4}$$

allowing more accurate steps thanks to the non-unit coefficient of $\mathbf{x}_u$, and enabling computationally efficient multistep solvers.

Several previous works highlight the importance of choosing the **timestep schedule** (the set of time points at which function evaluations are performed), which has a significant impact on the image generation quality (see (Karras et al., 2022, Appendix D.1) and (Frankel et al., 2025, Appendix H.3)).

## 2.3 SOLVER AND SCHEDULE DISTILLATION

Several recently introduced acceleration methods outsource the choice of solver coefficients and the timestep schedule to the gradient-based optimization. Specifically, LD3 (Tong et al., 2024) and S4S (Frankel et al., 2025) formulate this as an instance of knowledge distillation (Hinton et al., 2015). Given the pre-trained diffusion model and the corresponding ODE $d\mathbf{x}_t = \boldsymbol{v}(\mathbf{x}_t, t)dt$, one can define the complete "teacher" sampler to be the output of a multi-step high-quality approximation of the PF-ODE, which we denote by

$$\Phi^{\mathcal{T}}(\mathbf{x}_T) = \text{ODESolve}\left(\mathbf{x}_T, \boldsymbol{v}(\cdot, \cdot), T \to 0 \mid \text{Schedule, Solver; Params}\right). \tag{5}$$

Here, $\mathbf{x}_T$ is the initial value, $\boldsymbol{v}(\cdot, \cdot)$ is the corresponding velocity field and $T \to 0$ shows the interval, where we solve the ODE. "Solver" and "Schedule" define the sampling scheme and "Params" account

for the additional parameters of the scheme. Then, one could take any parameterization of the lightweight "student"

$$\Phi^{\mathcal{S}}(\mathbf{x}_T \mid \theta, \phi, \xi) = \text{ODESolve}\left(\mathbf{x}_T, \boldsymbol{v}(\cdot, \cdot), T \to 0 \mid \text{Schedule}(\theta), \text{Solver}(\phi); \text{Params}(\xi)\right) \quad (6)$$

with bounded computational requirements and optimize its parameters by minimizing a distance $\mathbf{d}$ between the corresponding outputs

$$\min_{\theta, \phi, \xi} \mathcal{L}_{\text{distill}}(\theta, \phi, \xi) = \min_{\theta, \phi, \xi} \mathbb{E}_{p_T(\mathbf{x}_T)} \mathbf{d}\left(\Phi^{\mathcal{S}}(\mathbf{x}_T \mid \theta, \phi, \xi); \ \Phi^{\mathcal{T}}(\mathbf{x}_T)\right). \quad (7)$$

In addition, LD3 and S4S account for the limited parameterization of the student and simplify its objective by allowing to slightly adapt the input and facilitate replication of the teacher output

$$\min_{\theta, \phi, \xi} \mathcal{L}_{\text{soft}}(\theta, \phi, \xi) = \min_{\theta, \phi, \xi} \mathbb{E}_{p_T(\mathbf{x}_T)} \min_{\mathbf{x}'_T \in \mathcal{B}(\mathbf{x}_T, r\sigma_T)} \mathbf{d}\left(\Phi^{\mathcal{S}}(\mathbf{x}'_T \mid \theta, \phi, \xi); \ \Phi^{\mathcal{T}}(\mathbf{x}_T)\right), \quad (8)$$

where $\mathcal{B}(\mathbf{x}_T, r\sigma_T) = \{\mathbf{x} : \|\mathbf{x} - \mathbf{x}_T\|^2 \leq r\sigma_T\}$ is the ball centered in $\mathbf{x}_T$ with a radius $r\sigma_T$ controlled by the additional hyperparameter $r$. We thoroughly discuss parameterizations of the methods and compare them with our Generalized Solver in Section 3.1.

## 2.4 ADVERSARIAL TRAINING

Adversarial training (Goodfellow et al., 2014) is a powerful way to guide a free-form generator $G_\theta(\mathbf{z})$ towards realistic outputs via optimizing the minimax objective (Nowozin et al., 2016)

$$\min_\theta \max_\psi \mathbb{E}_{p(\mathbf{z})} f\left(D_\psi(G_\theta(\mathbf{z}))\right) + \mathbb{E}_{p_{\text{data}}(\mathbf{x})} f\left(-D_\psi(\mathbf{x})\right). \quad (9)$$

Here, $f(t)$ is commonly equal to $-\log(1 + e^{-t})$, the discriminator $D_\psi$ is trained to distinguish real samples from the fake ones, while the generator aims to trick it. Family of the GAN losses with the form of Equation 9 (Nowozin et al., 2016; Mao et al., 2017; Lim & Ye, 2017) suffers from mode collapse (Arjovsky et al., 2017; Gulrajani et al., 2017). One of the alternatives is the relativistic GAN loss (Jolicoeur-Martineau, 2018)

$$\min_\theta \max_\psi \mathbb{E}_{p(\mathbf{z})p_{\text{data}}(\mathbf{x})} f\left(D_\psi(G_\theta(\mathbf{z})) - D_\psi(\mathbf{x})\right) \quad (10)$$

that is specifically designed to discourage mode dropping (Sun et al., 2020). Together with the gradient penalty

$$\mathcal{L}_{\text{grad}}(\theta, \psi) = \lambda_1 \mathbb{E}_{p_{\text{data}}(\mathbf{x})} \|\nabla_{\mathbf{x}} D_\psi(\mathbf{x})\|^2 + \lambda_2 \mathbb{E}_{p(\mathbf{z})} \|\nabla_{\mathbf{x}} D_\psi(G_\theta(\mathbf{z}))\|^2 \quad (11)$$

on discriminator outputs and architecture improvements, relativistic loss allows Huang et al. (2024) to build a novel high-quality GAN baseline R3GAN which we use throughout the paper.

## 3 METHOD

In this section, we construct **Generalized Adversarial Solver (GAS)**: an automatic sampler learning method that combines a simple yet effective parameterization with distillation and adversarial training.

## 3.1 GENERALIZED SOLVER (GS)

In Section 2.2 we have discussed that linear multi-step solvers and their specifically designed diffusion counterparts are the preferable families under strict requirements on computations. Given a timestep schedule $T = t_0 > t_1 > \ldots > t_N = \delta > 0$ and order $K$ they all have the same signature

$$\mathbf{x}_{n+1} = a_n \mathbf{x}_n + \sum_{j=\max(n-K+1, 0)}^{n} c_{j,n} \boldsymbol{v}(\mathbf{x}_j, t_j), \quad (12)$$

where the coefficients $a_n := a_n(t_n, t_{n+1})$ and $c_{j,n} := c_{j,n}(t_{j:n+1})$ typically depend on the current and the next timesteps. We propose several modifications to this basic signature. First, we stress that the less restriction on NFE is, the fewer parameters the method has. Second, one can see that depending on the parameterization of the diffusion model the formula may also contain the weighted

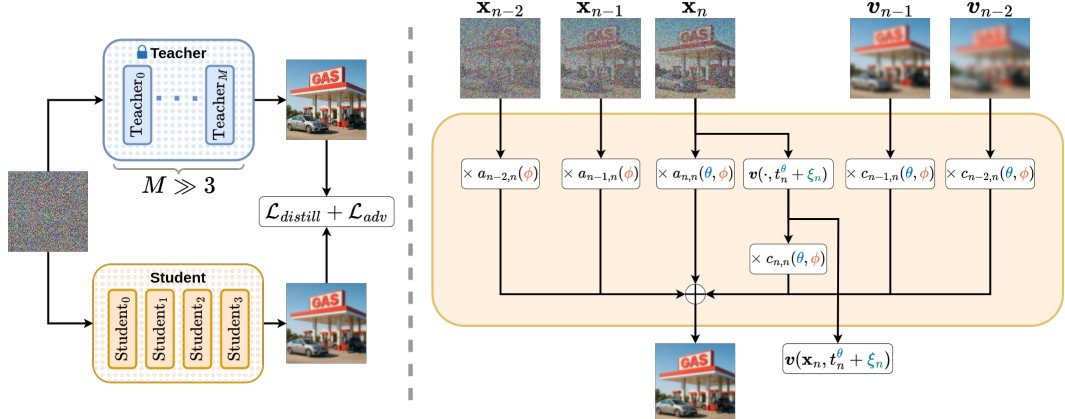

Figure 2: Illustration of the **Generalized Adversarial Solver**. Our student makes each sampling step by calculating the weighted average of all previous points and velocity directions. We train the corresponding weights and timestep schedule via distillation and adversarial loss.

sum of previous points (e.g., if one substitutes $\boldsymbol{v}(\mathbf{x}_j, t_j) = f(t_j)\mathbf{x}_j - (1/2)g^2(t_j)s(\mathbf{x}_j, t_j)$) along with the network predictions. We thus propose to increase the capacity of the signature by adding the weighted sum of all previous points [1] and remove the restriction on the order of the solver:

$$\mathbf{x}_{n+1} = \sum_{j=0}^{n} a_{j,n}\mathbf{x}_j + \sum_{j=0}^{n} c_{j,n}\boldsymbol{v}(\mathbf{x}_j, t_j). \tag{13}$$

Given this signature, we next define our parameterization that has three sets of parameters: $(\theta, \phi, \xi)$. **The first set $\theta$** of parameters defines the timestep schedule via the cumprod transformation: the logits $\theta_n$ are transformed into "stick breaking" portions $\sigma(\theta_n) \in [0, 1]$. The timesteps are then defined as

$$t_n^\theta = (T - \delta) \prod_{j=1}^{n} \sigma(\theta_j) + \delta. \tag{14}$$

**The second set $\phi$** defines the solver coefficients. However, we do not straightforwardly set $a_{j,n} := a_{j,n}(\phi)$ and $c_{j,n} := c_{j,n}(\phi)$. Instead, we use a powerful *base* multi-step solver (e.g. DPM-Solver++(3M) (Lu et al., 2022b)) as a source of *theoretical guidance* for the trained coefficients. This *base* solver offers time-dependent theoretical coefficients $a_{n,n}(t_{n:n+1}^\theta)$ and $c_{j,n}(t_{j:n+1}^\theta)$, which we can use as a strong backbone for our solver. We then train additive corrections to these coefficients in the following way. We set

$$a_{j,n}(\theta, \phi) := \begin{cases} a_{n,n}(t_{n:n+1}^\theta) + \hat{a}_{n,n}(\phi), & j = n; \\ \hat{a}_{j,n}(\phi), & \text{else,} \end{cases} \tag{15}$$

thus adding a trainable scalar $\hat{a}_{n,n}(\phi)$ to the current point coefficient $a_{n,n}(t_{n:n+1}^\theta)$ and training scalars $\hat{a}_{j,n}(\phi)$ for all the previous point coefficients.

Next, since the *"old"* velocities (computed more than $K$ steps before) do not have theoretical coefficients, we train one scalar $\hat{c}_{j,n}(\phi)$ per timestep $j \leq n - K$ and set

$$c_{j,n}(\theta, \phi) = \hat{c}_{j,n}(\phi). \tag{16}$$

Finally, we define the coefficients before the *"recent"* velocities (computed less than $K$ steps before). Here, theoretical base coefficients are typically constructed via weighted sum of the approximations $\hat{\boldsymbol{v}}_n^{(j)}$ of the higher-order derivatives $\boldsymbol{v}^{(j)}(\mathbf{x}_n, t_n^\theta)$ via finite differences (which are themselves weighted sums of previously computed velocities). This leads to the sum of the form $\sum_{j=0}^{K-1} \tilde{c}_{j,n}(t_{j:n+1}^\theta) \cdot \hat{\boldsymbol{v}}_n^{(j)}$.

---

[1]Theoretically, one could represent previous points as a linear combination of the previous velocity vectors. However, this "over-parameterization" may simplify training.

Table 1: Comparison of solver parameterizations between our GS, LD3 (Tong et al., 2024) and S4S (Frankel et al., 2025). We propose to add *additive guidance* to several velocity coefficients with a theoretical term from a pre-defined solver instead of just two multiplicative terms $a_n$ and $b_n$. The guidance is marked by the dependence of coefficients on $\theta$. We add a weighted sum of the previous points to the prediction.

| Method | Parameterization |
|---|---|
| LD3 | $\mathbf{x}_{n+1} = a_n(t_n^\theta, t_{n+1}^\theta) \cdot \mathbf{x}_n + \sum\limits_{j=\max(n-K+1,0)}^{n} c_{j,n}(t_j^\theta, \ldots, t_{n+1}^\theta) \cdot \boldsymbol{v}(\mathbf{x}_j, t_j^\theta + \xi_j)$ |
| S4S | $\mathbf{x}_{n+1} = a_n(t_n^\theta, t_{n+1}^\theta) \cdot \mathbf{x}_n + b_n(t_n^\theta, t_{n+1}^\theta) \cdot \sum\limits_{j=\max(n-K+1,0)}^{n} c_{j,n}(\phi) \cdot \boldsymbol{v}(\mathbf{x}_j, t_j^\theta + \xi_j)$ |
| **GS** | $\mathbf{x}_{n+1} = a_{n,n}(\theta, \phi) \cdot \mathbf{x}_n + \sum\limits_{j=0}^{n-1} a_{j,n}(\phi) \cdot \mathbf{x}_j + \sum\limits_{j=0}^{n} c_{j,n}(\theta, \phi) \cdot \boldsymbol{v}(\mathbf{x}_j, t_j^\theta + \xi_j)$ |

Combined with the finite-difference approximation of the derivatives $\hat{\boldsymbol{v}}_n^{(j)} = \sum_{i=n-j}^{n} \omega_{i,n} \cdot \boldsymbol{v}(\mathbf{x}_i, t_i^\theta)$, we obtain

$$\sum_{j=0}^{K-1} \tilde{c}_{j,n}(t_{j:n+1}^\theta) \cdot \sum_{i=n-j}^{n} \omega_{i,n} \cdot \boldsymbol{v}(\mathbf{x}_i, t_i^\theta). \tag{17}$$

Here, we train additive corrections $\hat{c}_{j,n}(\phi)$ for the coefficients $\tilde{c}_{j,n}(t_{j:n+1}^\theta)$ corresponding to the derivatives approximation. We thus obtain sum

$$\sum_{j=0}^{K-1} \left[ \tilde{c}_{j,n}(t_{j:n+1}^\theta) + \hat{c}_{j,n}(\phi) \right] \cdot \sum_{i=n-j}^{n} \omega_{i,n} \cdot \boldsymbol{v}(\mathbf{x}_i, t_i^\theta), \tag{18}$$

which produces *recent velocity coefficients*

$$c_{i,n}(\theta, \phi) = \omega_{i,n} \sum_{j=n-i}^{K-1} \left[ \tilde{c}_{j,n}(t_{j:n+1}^\theta) + \hat{c}_{j,n}(\phi) \right]. \tag{19}$$

We initialize the corrections with zeros to obtain an efficient initialization. By doing this, we ensure that even sudden change of the timesteps does not completely ruin the solver performance due to the meaningful dependence of its coefficients on time. We show the positive impact of the theoretical guidance in Section 4.2.

**The last set** $\xi$ of parameters acts as a correction to the timesteps that we evaluate the pre-trained model on. Analogous to Tong et al. (2024) and Frankel et al. (2025) we define the decoupled timesteps $t_j^\theta + \xi_j$ and use them for making predictions with the diffusion model. Combining the signature from Equation 13 with the introduced parameterization, we obtain the **Generalized Solver (GS)**

$$\mathbf{x}_{n+1} = a_{n,n}(\theta, \phi) \cdot \mathbf{x}_n + \sum_{j=0}^{n-1} a_{j,n}(\phi) \cdot \mathbf{x}_j + \sum_{j=0}^{n} c_{j,n}(\theta, \phi) \cdot \boldsymbol{v}(\mathbf{x}_j, t_j^\theta + \xi_j) \tag{20}$$

and extensively compare it with the parameterizations of LD3 and S4S in Table 1.

Taken together, the following design choices of the Generalized Solver improve its quality over existing approaches:

- the use of **theoretical coefficients**, which form the basis of GS and improve convergence;
- the signature of the **linear multistep method**, on which many theoretical solvers are based, determines the use of the past history of $\mathbf{x}_j$;
- **additive parameterization**, which connects the theoretical and trainable solver coefficients within the signature.

## 3.2 GENERALIZED ADVERSARIAL SOLVER (GAS)

We train the Generalized Solver on the previously established distillation loss from Equation 7. Specifically, we take $\mathbf{d}$ from the distillation loss (Equation 7) to be LPIPS in pixel-space and $L_1$ in latent-space experiments. We do not use the soft version from Equation 8. It is important to examine the "solver distillation" problem from another perspective. Essentially, it is an instance of the paired translation problem/learning a mapping from its input/output samples. Several works (Isola et al., 2017; Ledig et al., 2017) have shown that the standard regression loss could greatly benefit from adding the adversarial loss on the outputs. Recently, adversarial loss has been established as a powerful tool to boost performance of the diffusion distillation (Kim et al., 2023; Sauer et al., 2023; 2024; Yin et al., 2024) methods.

Given this, we augment distillation-based training of the GS via distillation loss and obtain the **Generalized Adversarial Solver (GAS)**. We denote our solver's output as

$$\Phi^{\mathcal{S}}\left(\mathbf{x}_T | \theta, \phi, \xi\right) = \mathrm{ODESolve}(\mathbf{x}_T, \boldsymbol{v}(\cdot, \cdot), T \to 0 \mid \mathrm{GS}(\theta, \phi, \xi)), \tag{21}$$

where $\mathrm{GS}(\theta, \phi, \xi)$ defines the Generalized Solver signature and parameterization, defined in Section 3.1 and Equation 20 specifically. We denote the discriminator by $D_\psi$ and train GAS on the sum of distillation and adversarial losses

$$\begin{cases} \min\limits_{\theta,\phi,\xi} \max\limits_{\psi} \mathcal{L}_{\mathrm{GAS}}(\theta, \phi, \xi, \psi) = \min\limits_{\theta,\phi,\xi} \max\limits_{\psi} \mathcal{L}_{\mathrm{distill}}(\theta, \phi, \xi) + \mathcal{L}_{\mathrm{adv}}(\theta, \phi, \xi, \psi); \\ \mathcal{L}_{\mathrm{adv}}(\theta, \phi, \xi, \psi) = \mathbb{E}_{p_T(\mathbf{x}_T)p_T(\mathbf{y}_T)} f\left(D_\psi\left(\Phi^{\mathcal{S}}\left(\mathbf{x}_T | \theta, \phi, \xi\right)\right) - D_\psi\left(\Phi^{\mathcal{T}}(\mathbf{y}_T)\right)\right). \end{cases} \tag{22}$$

We note that $\mathbf{x}_T$ and $\mathbf{y}_T$ are different initial noises for student and teacher generation sampled from the same prior distribution. We exploit R3GAN (Huang et al., 2024) relativistic loss with $f(t) = -\log(1 + e^{-t})$ and add the discriminator gradient penalties from Equation 11 to facilitate its training dynamics.

The incorporation of the adversarial loss is also effective in terms of removing generation artifacts in low NFE regimes, where regression task becomes harder. We will further demonstrate this in Section 4.

## 4 EXPERIMENTS

We demonstrate the efficiency of the proposed method by conducting experiments on several pixel and latent space experiments. We perform evaluation on pixel-space CIFAR10 (32×32) (Krizhevsky & Hinton, 2009), FFHQ (64×64) (Karras et al., 2019), and AFHQv2 (64×64) (Choi et al., 2020). Among latent diffusion models (Rombach et al., 2022) we cover LSUN Bedroom (256×256) (Yu et al., 2015) and the class-conditional ImageNet (256×256) (Russakovsky et al., 2015). Additionally, we assess the Stable Diffusion (Rombach et al., 2022) model on the MSCOCO (512×512) (Lin et al., 2015) text-to-image dataset. We use Karras et al. (2022) and Rombach et al. (2022) pretrained models for pixel and latent space experiments respectively.

We choose distance $\mathbf{d}$ (Equation 7) in distillation loss to be LPIPS (Zhang et al., 2018) in pixel-space and $L_1$ in latent-space experiments. We initialize timesteps using a time-uniform schedule and utilize the DPM-Solver++(3M) (Lu et al., 2022b) coefficients as the guiding theoretical parameters. For pixel-space models we use a pretrained R3GAN discriminator. For latent experiments we adapt the same discriminator architecture, but train it from scratch. We calculate FID (Heusel et al., 2017) using 50000 samples, unless stated otherwise. The additional training details can be found in Appendix D.

## 4.1 MAIN RESULTS

In Table 2 we illustrate that the proposed methods, GS and GAS, systematically enhance image sampling quality across different solvers, especially in low NFE setups. As an example, the S4S Alt (Frankel et al., 2025) algorithm reports a FID score of 10.63 with NFE=4 on the FFHQ dataset, whereas GAS achieves a significantly better FID score of **7.86** under the same conditions. Our approach outperforms all previously proposed methods across all evaluated datasets. Specifically, GAS achieves a FID score of **4.48** with NFE=4 on the AFHQv2 dataset and **3.79** on the FFHQ dataset using NFE=6. Additionally, we achieve the FID score of **5.38** on the conditional ImageNet dataset

with NFE=4, **4.60** on the LSUN Bedrooms dataset with NFE=5, and **14.71** on the MS-COCO dataset with NFE = 4.

Table 2: We evaluate FID score comparison of the proposed GS and GAS methods against existing solvers like UniPC and iPNDM, and alongside training-based approaches such as GITS, DMN, LD3, and S4S. We report the FIDs of the teacher models as those utilized during our training process. The baseline scores were taken from the corresponding papers, unless otherwise noted.† report utilizing teacher model having significantly difference in teacher hyperparameters, thus it cannot be fairly compared to other methods.

(a) Pixel-space datasets include CIFAR10 ($32 \times 32$), AFHQv2 ($64 \times 64$), and FFHQ ($64 \times 64$)

| Method | NFE=4 | NFE=6 | NFE=8 | NFE=10 |
|---|---|---|---|---|
| **CIFAR10** | | | | |
| *Solvers* | | | | |
| DPM++ (3M) | 46.59 | 12.16 | 4.62 | 3.08 |
| UniPC (3M) | 43.92 | 13.12 | 4.41 | 3.16 |
| iPNDM (3M) | 35.04 | 11.80 | 5.67 | 3.69 |
| *Solver optimization methods* | | | | |
| UniPC [GITS] | 25.32 | 11.19 | 5.67 | 3.70 |
| UniPC [DMN] | 26.35 | 8.09 | 5.90 | 2.45 |
| iPNDM [GITS] | 15.63 | 6.82 | 4.29 | 2.78 |
| iPNDM [DMN] | 28.09 | 9.24 | 7.68 | 3.31 |
| Best LD3 | 9.31 | 3.35 | 2.81 | 2.38 |
| S4S Alt | 6.35 | 2.67 | 2.39 | 2.18 |
| GS (Ours) | 4.41 | 2.55 | 2.25 | 2.18 |
| GAS (Ours) | **4.05** | **2.49** | **2.24** | **2.17** |
| Teacher | | 2.03 | | |
| **FFHQ** | | | | |
| *Solvers* | | | | |
| DPM++ (3M) | 46.14 | 14.01 | 6.18 | 4.18 |
| UniPC (3M) | 53.25 | 11.24 | 5.59 | 3.90 |
| iPNDM (3M) | 36.54 | 16.44 | 8.11 | 5.39 |
| *Solver optimization methods* | | | | |
| UniPC [GITS] | 21.38 | 12.21 | 7.84 | 4.46 |
| UniPC [DMN] | 25.82 | 9.47 | 6.85 | 3.54 |
| iPNDM [GITS] | 18.05 | 9.38 | 5.72 | 3.96 |
| iPNDM [DMN] | 31.30 | 12.12 | 11.00 | 5.24 |
| Best LD3 | 17.96 | 5.97 | 3.50 | 3.25 |
| S4S Alt | 10.63 | 4.62 | 3.15 | 2.91 |
| GS (Ours) | 10.70 | 4.49 | 2.96 | 2.67 |
| GAS (Ours) | **7.86** | **3.79** | **2.87** | **2.66** |
| Teacher | | 2.60 | | |
| **AFHQv2** | | | | |
| *Solvers* | | | | |
| DPM++ (3M) | 27.82 | 10.72 | 4.28 | 3.19 |
| UniPC (3M) | 33.78 | 8.27 | 4.60 | 3.81 |
| iPNDM (3M) | 23.20 | 9.55 | 4.49 | 3.19 |
| *Solver optimization methods* | | | | |
| UniPC [GITS] | 12.20 | 7.26 | 3.86 | 2.88 |
| UniPC [DMN] | 30.32 | 14.46 | 6.85 | 2.94 |
| iPNDM [GITS] | 12.89 | 6.10 | 4.03 | 3.26 |
| iPNDM [DMN] | 33.15 | 16.01 | 10.12 | 3.22 |
| Best LD3 | 9.96 | 3.63 | 2.63 | 2.27 |
| S4S Alt | 6.52 | 2.70 | **2.29** | **2.18** |
| GS (Ours) | 5.92 | 2.87 | 2.33 | 2.25 |
| GAS (Ours) | **4.48** | **2.66** | **2.29** | 2.31 |
| Teacher | | 2.16 | | |

(b) Latent diffusion models are tested on the LSUN-Bedroom and ImageNet datasets ($256 \times 256$).

| Method | NFE=4 | NFE=5 | NFE=6 | NFE=7 |
|---|---|---|---|---|
| **LSUN-Bedroom-256 (latent space)** | | | | |
| *Solvers* | | | | |
| DPM++ (3M) | 48.82 | 18.64 | 8.50 | 5.16 |
| UniPC (3M) | 39.78 | 13.88 | 6.57 | 4.56 |
| iPNDM (3M) | 11.93 | 6.38 | 5.08 | 4.39 |
| *Solver optimization methods* | | | | |
| UniPC [GITS] | 70.93 | 47.37 | 22.33 | 17.27 |
| UniPC [DMN] | 29.22 | 8.21 | 4.40 | 4.55 |
| iPNDM [GITS] | 76.86 | 59.17 | 28.09 | 19.54 |
| iPNDM [DMN] | 11.82 | 6.15 | 4.71 | 5.16 |
| Best LD3 | 8.48 | 5.93 | 4.52 | 4.16 |
| S4S Alt† | 20.89 | 13.03 | 10.49 | 10.03 |
| GS (Ours) | 9.83 | 5.32 | **3.77** | **3.34** |
| GAS (Ours) | **6.68** | **4.60** | **3.77** | 3.36 |
| Teacher | | 3.06 | | |
| **Imagenet-256 (latent space)** | | | | |
| *Solvers* | | | | |
| DPM++ (3M) | 26.07 | 11.91 | 7.51 | 5.95 |
| UniPC (3M) | 20.01 | 8.51 | 5.92 | 5.20 |
| iPNDM (3M) | 13.86 | 7.80 | 6.03 | 5.35 |
| *Solver optimization methods* | | | | |
| UniPC [GITS] | 54.88 | 34.91 | 14.62 | 9.04 |
| UniPC [DMN] | 16.72 | 7.96 | 7.54 | 7.81 |
| iPNDM [GITS] | 56.00 | 43.56 | 19.33 | 10.33 |
| iPNDM [DMN] | 10.15 | 7.33 | 7.25 | 7.40 |
| Best LD3 | 9.19 | 5.03 | 4.46 | 4.32 |
| S4S Alt† | **5.13** | **4.30** | **4.09** | **4.06** |
| GS (Ours) | 7.87 | 4.93 | 4.30 | 4.17 |
| GAS (Ours) | 5.38 | 4.87 | 4.32 | 4.17 |
| Teacher | | 4.10 | | |

(c) Training dataset for SD consists of 1000 MS-COCO samples, while FID is computed across 30,000 prompts to generate images with spatial resolution of $512 \times 512$.

| Method | NFE=4 | NFE=5 | NFE=6 | NFE=7 |
|---|---|---|---|---|
| **MS-COCO (Stable Diffusion v1.5)** | | | | |
| iPNDM (2M) | 17.76 | 14.41 | 13.86 | 13.76 |
| iPNDM [GITS] | 18.05 | 14.11 | 12.10 | 11.80 |
| Best LD3 | 17.32 | 13.07 | 12.40 | 11.83 |
| S4S† | 16.05 | 13.26 | **11.17** | **10.83** |
| GS (Ours) | 14.94 | 11.97 | 11.71 | 11.32 |
| GAS (Ours) | **14.71** | **11.91** | 11.73 | 11.36 |
| Teacher | 14.10 | 12.08 | 11.80 | 11.48 |

## 4.2 ABLATION STUDY

**Coefficients parametrization**  First, we demonstrate significant impact of solver parameterization on training efficiency. Specifically, we show the difference between our parameterization, that represents coefficients as sum of fixed theoretical guidance and explicitly trained additive corrections,

Table 3: We compare our parametrization with S4S variant on CIFAR10 and FFHQ datasets in terms of FID and LPIPS scores. Both setups use batch size of 24, while training dataset consists of 49k samples. Teacher dataset has FID score of 2.03 and 2.60 for CIFAR10 and FFHQ datasets respectively.

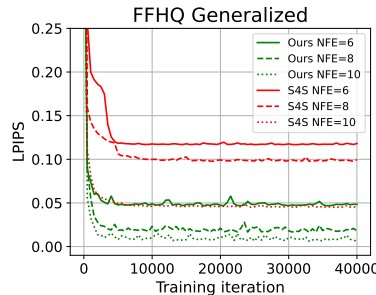

|  | NFE=4 | | NFE=6 | | NFE=8 | | NFE=10 | |
|---|---|---|---|---|---|---|---|---|
|  | FID | LPIPS | FID | LPIPS | FID | LPIPS | FID | LPIPS |
| **CIFAR10** | | | | | | | | |
| S4S | 31.44 | 0.273 | 2.93 | 0.073 | 2.87 | 0.072 | 2.26 | 0.027 |
| Our | **4.39** | **0.116** | **2.51** | **0.046** | **2.21** | **0.017** | **2.15** | **0.010** |
| **FFHQ** | | | | | | | | |
| S4S | 24.24 | 0.175 | 11.08 | 0.117 | 7.76 | 0.098 | 3.97 | 0.045 |
| Our | **10.79** | **0.116** | **4.40** | **0.046** | **2.97** | **0.016** | **2.70** | **0.005** |

Figure 3: LPIPS evaluation loss for training iterations comparing S4S and our parametrization. Our method results in more stable training process.

and the parameterization from another high-quality method S4S (Frankel et al., 2025). We ablate the theoretical guidance in Appendix B.1 and demonstrate that it yields a substantial improvement in FID. For the purpose of a fair comparison with S4S, we implemented LMS + PC S4S solver type removing a constraint on the solver order. This guarantees that Generalized Solver and S4S have the same number of trainable parameters.

In Table 3 we demonstrate our parameterization's superior performance on different datasets and NFE. Our results are consistent with the training issue reported in (Frankel et al., 2025). Figure 3 represents the dynamics of LPIPS loss on the evaluation dataset in different training iterations of the experiment. Our parametrization shows a more efficient training process, faster convergence and more stable training behavior. We also compare GS with LMS and LMS+PC under identical configurations in Appendix B.9.

We observe that training with our parametrization for GS and GAS is stable, demonstrating an improvement in FID throughout the training process. More details are provided in Appendix B.3.

**Adversarial training** Addition of the adversarial training is a crucial part of our contribution, because it significantly improves the image generation quality as seen in Tables 2a, 2b. It is crucial for low NFE setups because a teacher image can be too difficult for the student to replicate, therefore smaller values of the regression loss (LPIPS or $L_1$ for pixel and latent models respectively) do not always correlate with smaller FID scores (as can be seen in Table 4) and occasionally result in visible artifacts. Examples of such behavior are presented in Figure 4. Adding adversarial loss makes the student's generation closer to teacher's distribution and thus removes appearing artifacts and makes generation more realistic, in spite of occasionally resulting in bigger LPIPS or L1 losses. Additionally, we discuss three aspects of adversarial training in the appendices: the influence of loss selection in Appendix B.2, the sensitivity to the adversarial loss weight in Appendix B.5 and the impact of the training itself on mode collapse in Appendix B.6.

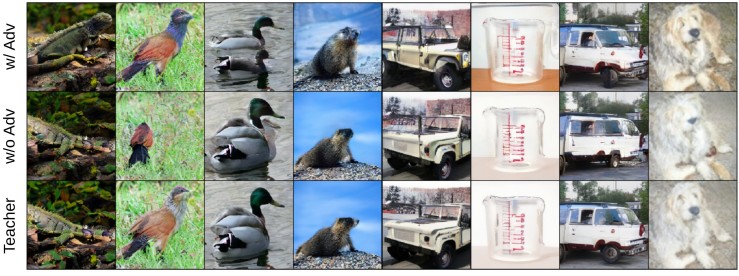

Figure 4: Incorporating an adversarial loss into the training process enhances generation quality, reducing occurring image artifacts in low NFE regimes. In this setup, the teacher model uses UniPC (3M) solver with NFE=10, while the student models operate with a reduced NFE=4.

Table 4: Results of 10k training iterations calculated on 1000 validation samples.

|  | FID | $\mathcal{L}_{distill}$ |
|---|---|---|
| **FFHQ** | | |
| GS | 10.70 | **0.116** |
| GAS | **7.86** | 0.127 |
| **LSUN** | | |
| GS | 9.64 | **0.172** |
| GAS | **7.54** | 0.174 |

### 4.3 METHOD EFFICIENCY

We next show that GAS is efficient in terms of dataset size and training time.

**Dataset size**  By the dataset we mean the set of samples generated from the teacher model to use in the training process. To this end, we measure method's performance on the "full" dataset scenario with 49000 samples and find the smaller dataset size that demonstrates equivalent results. First, we observe that the dataset size of 1400 is enough for training GS without adversarial loss. However, the solver's optimization problem becomes more challenging in low-NFE scenarios with adversarial loss. Here, we expand the dataset from 1400 samples to 5000 and obtain results indistinguishable from the full-dataset scenario in all datasets and settings. Additional information is provided in Appendix C.1.

**Performance**  Without adversarial training, GS converges within 1-2.5 hours depending on the dataset, which is comparable to the most relevant baselines LD3 and S4S. In case of GAS, training time increases to 2-9 hours, which is larger, but still requires similar order. We refer the reader to the Appendix C.2 for the exact comparison of metrics depending on training time and Appendix C.3 for peak-memory usage in the backward pass.

## 5 DISCUSSION

In this paper, we propose Generalized Adversarial Solver, the novel parameterization and training algorithm for automatic gradient-based solver optimization. The main novelty is additive theoretical guidance of solver coefficients and combination of distillation loss with adversarial training. We establish that the introduced Generalized Solver parameterization significantly outperforms existing parameterizations. We show that adding the adversarial loss significantly boosts method's performance and allows tackling the image artifacts present in simple solver distillation. We extensively compare our method with other solver/timestep training approaches and demonstrate its superior performance on 6 datasets, ranging from $32 \times 32$ pixel-space CIFAR10 to $256 \times 256$ latent-space ImageNet and $512 \times 512$ MS-COCO with Stable Diffusion.

**Limitations**  Our method relies on performing backpropagation through the whole solver inference, which may face scalability issues when applied to larger image sizes and bigger models. We explore the generalizability of our method between different datasets in Section B.4. However, a potential concern remains as to whether GS/GAS requires separate training for each preferred inference NFE. We leave the development of lightweight modifications to our method for future work.

The weights of the pretrained diffusion model remain frozen during solver training. We compare GS and GAS with distillation-based methods and show in Appendix B.7 that our approach better preserves the generative capabilities of the original model. However, due to the limited number of trainable parameters at NFE=1 and 2, our method yields lower quality in these specific settings; perfomance GS and GAS at low NFE is provided in Appendix B.8.

### REPRODUCIBILITY STATEMENT

To ensure the clarity and reproducibility of our work, we provide excessive description of all parts of our method. Appendix D provides the pseudocode of our algorithm, exactly matching the way it appears in our implementation; configurations and hyperparameters of all "teacher" generations and "student" training processes, including batch sizes, optimizer choice and other fine-grained details; and expressions for commonly used timestep schedules mentioned in the paper.

Furthermore, our experiments are built upon publicly available datasets (e.g., CIFAR10, FFHQ) and pre-trained model checkpoints to ensure our experimental setups are accessible and verifiable.

### ACKNOWLEDGMENTS

Aleksandr Oganov wishes to express sincere gratitude to his alma mater, Lomonosov Moscow State University, for providing the foundational education and stimulating environment that made this research possible. Mishan Aliev thanks Yandex Education for supporting him during this research.

We thank Dmitry Baranchuk for fruitful discussions and valuable advice on knowledge distillation techniques. This research was supported in part through computational resources of HPC facilities at HSE University. A special thanks to Maxim Kodryan — his mere existence was contribution enough. The work was supported by the grant for research centers in the field of AI provided by the Ministry of Economic Development of the Russian Federation in accordance with the agreement 000000C313925P4E0002 and the agreement with HSE University №139-15-2025-009. We are thankful for the ICLR reviewers for their detailed suggestions that led to a significant improvement of this paper.

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

## A   RELATED WORK

Among many **inference-time** acceleration algorithms, solver-based methods treat diffusion models as ODEs with a (partially) black-box velocity function. Specifically, PNDM (Liu et al., 2022a) and iPNDM (Zhang & Chen, 2022) apply the linear multistep method to the corresponding PF-ODE. DPM-Solver (Lu et al., 2022a), DEIS (Zhang & Chen, 2022) use the variation of constants (Equation 4) and approximate the underlying integral. DPM-Solver++ (Lu et al., 2022b) extends this idea to the multi-step version, and UniPC (Zhao et al., 2024) modifies it with the predictor-corrector framework. Besides the solver distillation loss, introduced for optimizing the timesteps in LD3 (Tong et al., 2024) and used for optimizing both timesteps and solver coefficients in S4S (Frankel et al., 2025), many automatic solver selection methods were proposed. DDSS (Watson et al., 2021) directly optimizes generation quality of the solver. AYS (Sabour et al., 2024) optimizes timesteps to minimize the KL divergence between the backward SDE and the discretization. GITS (Chen et al., 2024a) choose the timesteps by utilizing trajectory structure of the PF-ODE and DMN (Xue et al., 2024) allows for the fast model-free choice of parameters via optimizing an upper-bound on the solution error. Some approaches manipulate **diffusion-specific properties** and utilize redundancies in their computations. Namely, DeepCache (Ma et al., 2024) and CacheMe (Wimbauer et al., 2024) propose to perform block or layer caching and reuse activations from the previous timesteps. The other directions of acceleration include quantization (Gu et al., 2022; Badri & Shaji, 2023) and pruning (Fang et al., 2023; Castells et al., 2024).

In contrast, **diffusion distillation** techniques aim at compressing a pre-defined diffusion model by training a few-step student. Several methods learn to mimic solution of the PF-ODE. This includes optimizing the regression loss between the outputs (Salimans & Ho, 2022) or learning the integrator between arbitrary timesteps (Gu et al., 2023; Song et al., 2023; Kim et al., 2023). Others use diffusion models as a training signal that assesses likelihood of the generated images. It is commonly formalized as optimizing the Integrated KL divergence (Luo et al., 2024; Yin et al., 2023; 2024; Nguyen & Tran, 2023) by training an additional "fake" diffusion model on the generator's output distribution. Other methods consider matching scores (Zhou et al., 2024a) or moments (Salimans et al., 2024) of the corresponding distributions. Many distillation methods enhance student generation quality by adding the adversarial training (Kim et al., 2023; Yin et al., 2024), including discriminator loss on detector (Sauer et al., 2023) or teacher features (Sauer et al., 2024).

## B   ADDITIONAL EXPERIMENTS

### B.1   THEORETICAL GUIDANCE FOR THE COEFFICIENTS

The integration of theoretical guidance for the solver coefficients is important for the effective training of GS. It provides a strong inductive bias by embedding knowledge of the theoretically optimal coefficients directly into GS. Thus, in the optimization process, the trained coefficients do not have to learn complex theoretical dependencies, since they are already embedded in the theoretical coefficients.

To empirically validate the contribution, we conducted the ablation by training a version of GS on the FFHQ dataset both with and without this guidance. In both configurations we use hyperparameters from D.4.1, the coefficients were initialized to be equivalent to those of DPM-Solver++(3M). The results, presented in Table 5, demonstrate that the inclusion of theoretical guidance yields a substantial improvement in FID. Furthermore, the flexible signature of GS enables the generalization of various modern theoretical solvers, opening up new avenues for research into different forms of theoretical guidance.

### B.2   ADVERSARIAL LOSS

To better understand the impact of adversarial loss, we compared different GAN losses on ImageNet at NFE=4. We compared the standard GAS, which uses the relativistic loss from Equation 10, with one trained using the traditional loss from Equation 9. In both configurations, we used discriminator gradient penalties and the same hyperparameters from Appendix D.4.2.

Table 5: GS with and without theoretical guidance on FFHQ dataset in terms of FID. Both setups use same hyperparameters and the coefficients were initialized to be equivalent to those of DPM-Solver++(3M).

| Parameterization | NFE=4 | NFE=6 | NFE=8 | NFE=10 |
|---|---|---|---|---|
| w/ theory | **10.70** | **4.49** | **2.96** | **2.67** |
| w/o theory | 15.23 | 10.53 | 5.50 | 4.69 |

Table 6: GAS with traditional and relativistic GAN losses on ImageNet dataset at NFE=4 in terms of FID.

| GS | GAS (traditional) | GAS (relativistic) |
|---|---|---|
| 7.87 | 6.49 | **5.38** |

From the comparison results presented in Table 6, we conclude that employing a GAN loss enhances the final output quality. GAS demonstrates strong performance with both traditional and relativistic losses. Although the relativistic loss is optional, it leads to a superior model.

### B.3    FID PROGRESSION DURING TRAINING

To better understand the training process, we visualize the dynamics of the FID score during the training process.

When comparing the GS and GAS FID scores for FFHQ, as visualized in Figure 5a, we observe that incorporating the adversarial objective requires more training iterations for our method to converge. However, it is more important that, as previously reported in Table 2a, it achieves a significantly lower FID score, allowing for a better trade-off between generation quality and a slight increase in training time.

Figure 5b demonstrates that although GAS achieves excellent FID scores after 30k iterations, it could potentially yield even better results with further training. This is suggested by the continuing decrease in the FID score for NFE of 4 and 5 with each additional training iteration. Scenarios involving a larger number of NFE for model inference do not display this pattern, since they comprise a bigger student's capacity and lead to easier optimization task and earlier convergence.

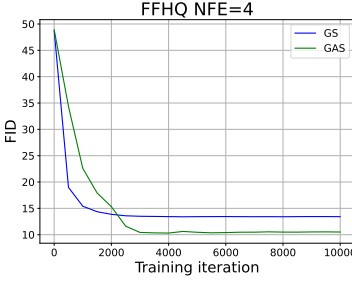

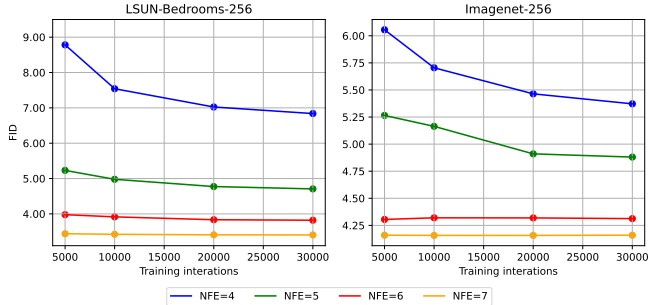

(a) FID values during training for the FFHQ dataset, using 4 NFE with the **GS** and **GAS**. We evaluate FID every 500 training iterations, computing it based on 5000 generated samples.

(b) **GAS** FID training dynamics for latent space datasets for several NFE scenarios. We generate 50k images for each of 5000, 10k, 20k and 30k iterations checkpoints, and evaluate FID scores based on those datasets.

Figure 5: FID score for several checkpoints during training of GS and GAS for both pixel (FFHQ) and latent space (LSUN, ImageNet) datasets.

### B.4    GENERALIZATION ACROSS DATASETS

Regarding generalization across datasets with significantly different dimensionalities (e.g., CIFAR vs. COCO), the optimal schedule for a smaller resolution may not be optimal for higher resolutions due to simpler denoising tasks at equivalent noise levels (larger images have greater correlation among nearby pixels). To further demonstrate the method's generalization results, we tested solver transfer between closely related diffusion models (FFHQ and AFHQv2), demonstrating practical generalizability. We thus illustrate its generalization in Table 7.

Table 7: We evaluate FID score comparison of GS and GAS trained on dataset and applied on another against DPM-Solver++, LD3 and S4S. We use the GS and GAS checkpoints as in Table 2a.

(a) Solvers GS, GAS, trained on FFHQ and applied on AFHQv2 (denoted as GS' and GAS') consistently outperform baseline methods.

| Method | NFE=4 | NFE=6 | NFE=8 | NFE=10 |
|---|---|---|---|---|
| DPM-Solver++ | 27.82 | 10.72 | 4.28 | 3.19 |
| Best LD3 | 9.96 | 3.63 | 2.63 | 2.27 |
| S4S Alt | 6.52 | 2.70 | 2.29 | 2.18 |
| GS (Ours) | 5.92 | 2.87 | 2.33 | 2.25 |
| GAS (Ours) | 4.48 | 2.66 | 2.29 | 2.31 |
| GS' (Ours) | 6.54 | 3.01 | 2.41 | 2.29 |
| GAS' (Ours) | 5.15 | 2.81 | 2.44 | 2.32 |

(b) Solvers GS, GAS, trained on AFHQv2 and applied on FFHQ (denoted as GS' and GAS') consistently outperform baseline methods.

| Method | NFE=4 | NFE=6 | NFE=8 | NFE=10 |
|---|---|---|---|---|
| DPM-Solver++ | 46.14 | 14.01 | 6.18 | 4.18 |
| Best LD3 | 17.96 | 5.97 | 3.50 | 3.25 |
| S4S Alt | 10.63 | 4.62 | 3.15 | 2.91 |
| GS (Ours) | 10.70 | 4.49 | 2.96 | 2.67 |
| GAS (Ours) | 7.86 | 3.79 | 2.87 | 2.66 |
| GS' (Ours) | 16.01 | 5.91 | 3.27 | 2.70 |
| GAS' (Ours) | 9.39 | 4.21 | 2.92 | 2.72 |

## B.5 ADVERSARIAL LOSS WEIGHT

One of the few hyperparameters of GAS is the GAN-weight. Starting from the resolution of $64 \times 64$, the weight of the adversarial loss was fixed to 1.0 for all datasets. Figure 6 demonstrates that GAS is insensitive to the GAN-weight selection and achieves similar FID with different weights. This shows that our method achieves strong results without the need for hyperparameter tuning.

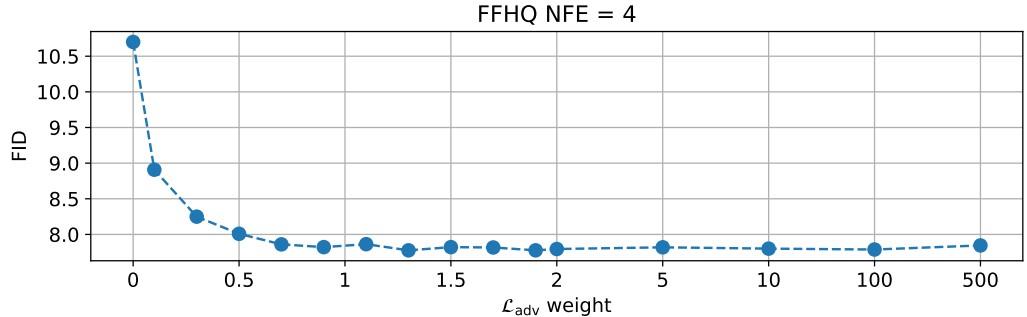

Figure 6: FID values for FFHQ dataset with 4 NFE for different adversarial loss weights. The metric remains stable even at large weight values. Setting $\mathcal{L}_{\text{adv}} = 0$ for GAS results in absence of adversarial training, thus is a GS setting.

## B.6 MODE COLLAPSE

To demonstrate that GAS maintains diversity, we computed precision, recall, density, coverage from Naeem et al. (2020) for GS and GAS, comparing their statistics to those of the teacher model. Evaluation results were obtained by comparing 50000 generated samples to 50000 teacher images using T4096(Naeem et al. (2020), Section 4.2). We report results on the ImageNet dataset with NFE=4 in Table 8. We choose this setup because the incorporation of the adversarial loss was significant in this experiment and could raise the greatest concerns regarding mode-collapse. As shown in Table 8, GAS does not suffer from the aforementioned problem and even increases mode coverage and recall compared to GS, which is trained without adversarial loss. Additionally, we provide random samples in Appendix F.

Table 8: Mode collapse ablation of the GAS on ImageNet dataset.

| Method | FID | precision | recall | density | coverage |
|---|---|---|---|---|---|
| GS | 7.87 | **0.90** | 0.63 | 1.18 | 0.90 |
| GAS | **5.38** | **0.90** | **0.76** | **1.20** | **0.97** |

Table 9: Comparison of generative properties of models on CIFAR10 dataset.

| Method | CD, NFE=1 | CD, NFE=4 | GS, NFE=4 | GAS, NFE=4 | Our teacher |
|---|---|---|---|---|---|
| Coverage | 0.942 | 0.938 | 0.963 | 0.961 | 0.971 |
| FID | 3.56 | 2.99 | 4.41 | 4.05 | 2.03 |

### B.7 Comparison with Consistency Distillation

We compare GS and GAS with distillation-based methods in terms of their ability to preserve the generative properties of the diffusion model. To evaluate diversity, we compare the coverage from Naeem et al. (2020) and FID of our methods (GS and GAS) against the official checkpoint of CD Song et al. (2023) on CIFAR10. We assess generation quality against a teacher solver, which achieves FID=2.03 and coverage=0.971. Throughout, when working with CD, we used the official implementation and ternary search for NFE=4(Song et al. (2023), Section 3). Coverage was measure between 10000 generated images and the CIFAR10 test set using T4096.

The comparison results for FID and coverage are presented in Table 9 and show that GS and GAS have higher coverage than CD. In addition, CD demonstrates low coverage compared to the teacher solver, which indicates a deterioration in the generative properties of the diffusion model. GS and GAS, without changing these parameters, provide a more flexible and property-preserving acceleration method. We compare the efficiency of GS and GAS with that of distillation-based approaches in Appendix C.2.

### B.8 Performance at low NFE

We evaluated the performance of GS and GAS at low NFE. We trained GS and GAS on the FFHQ dataset with NFE=1 and NFE=2. For training, we used the hyperparameters from Appendix D.4.1. For comparison, we used DPM-Solver with order set to NFE when NFE < 4, and order=3 when NFE=4. Table 11 shows that GS and GAS improve upon DPM-Solver, but they do not perform well at low NFE as the FID scores are too high.

### B.9 Coefficients parametrization

In addition to the coefficient parameterization ablation in Section 4.2, we compared GS with the S4S LMS and LMS+PC parameterizations under the same hyperparameters and within our codebase (S4S does not provide an official implementation). In Table 3, we compare GS with LMS+PC when all methods are trained to convergence. Here, we further show that GS outperforms both LMS and LMS+PC on FFHQ under identical training settings from Appendix D.4.1. We initialize LMS and GS so that the corresponding solvers are initially equivalent to DPM-Solver++(3M). Table 10 shows that for all NFE values, GS achieves better generation quality.

Table 10: Comparison of S4S parameterizations with GS on FFHQ dataset in terms of FID. We use same hyperparameters, teachers and codebase.

| Parameterization | NFE=4 | NFE=6 | NFE=8 | NFE=10 |
|---|---|---|---|---|
| S4S LMS | 17.05 | 5.93 | 4.37 | 3.96 |
| S4S LMS + PC | 45.51 | 24.04 | 7.95 | 4.01 |
| GS | **10.70** | **4.49** | **2.96** | **2.67** |

Table 11: Comparison of GS and GAS with DPM-Solver++ on FFHQ dataset in terms of FID. We denoted DPM as DPM-Solver++.

| | NFE=1 | NFE=2 | NFE=4 |
|---|---|---|---|
| DPM | 314.95 | 134.36 | 46.14 |
| GS | **147.54** | **55.98** | 10.70 |
| GAS | 193.22 | 60.69 | **7.86** |

## C Efficiency of the method

### C.1 Training dataset size

We conduct experiments to assess the efficiency of the proposed methods with respect to the size of the training dataset. We examine several variations of sizes: 49000 as a baseline, 5000 and 1400 as the more lightweight alternatives. For GS, we observe that taking 1400 images and performing 10000 training iterations is sufficient for our method to converge, regardless of NFE. We note that it reaches equivalent or better FID scores compared to a bigger training dataset (see Table 12a).

The same pattern occurs with GAS on CIFAR10. The dataset of 1400 images is optimal for its training. However, starting from the higher-dimensional FFHQ dataset, we observe the typical challenges of adversarial training. As the discriminator used in GAS is trained simultaneously with

the other parameters of the solver, it tends to overfit and demands larger dataset size to alleviate this problem.

Adversarial training has demonstrated its effectiveness, especially in scenarios with smaller inference steps. We thus illustrate its performance in Table 12b on NFE = 4 and NFE = 6. It shows that the training dataset size of 5000 is sufficient for matching performance of the model trained on 49000.

Table 12: Comparison of different dataset sizes with several NFE, where N indicates the number of samples in the training dataset. In Table 12a the FID score is calculated after 10k and 20k training iterations to show the early convergence of the GS method. Table 12b presents the results of GAS evaluation after 10k iterations of training.

(a) Generalized solver

| | | NFE=4 | | NFE=10 | |
|---|---|---|---|---|---|
| | N | 10k | 20k | 10k | 20k |
| CIFAR10 | 1400 | 4.35 | 4.35 | 2.14 | 2.15 |
| | 49000 | 4.39 | 4.39 | 2.15 | 2.15 |
| FFHQ | 1400 | 10.70 | 10.72 | 2.71 | 2.71 |
| | 49000 | 10.79 | 10.82 | 2.70 | 2.71 |

(b) Generalized Adversarial solver

| | N | NFE=4 | NFE=6 |
|---|---|---|---|
| CIFAR10 | 1400 | 3.98 | 2.44 |
| | 49000 | 3.98 | 2.48 |
| FFHQ | 1400 | 9.44 | 4.48 |
| | 5000 | 7.83 | 3.79 |
| | 49000 | 7.93 | 3.76 |

## C.2 TRAINING TIME

We further investigate GS/GAS training dynamics by estimating their convergence time and comparing their computational efficiency with other methods.

In Table 13a we demonstrate the training time of Progressive Distillation (PD, (Salimans & Ho, 2022)) and Consistency Distillation (CD, (Song et al., 2023)). Those methods focus on training a new generator model that can sample images in a few-NFE manner. Both require days of training time and are computationally demanding.

We also compare our methods with several approaches that involve training certain parameters of solvers. **In pixel space** GS requires less than an hour of training time on CIFAR10, which is comparable to LD3, S4S and S4S-Alt. Notably, it achieves FID of 2.44 with NFE = 6, while S4S-Alt results in FID score of 2.52 with NFE = 7 and equivalent training time. Adversarial loss extends the training time to up to 2 hours, however, as we report in Table 2a, it achieves superior results in terms of FID score.

In the **latent** diffusion setting, we compare our method with LD3, which reports convergence within an hour of training time. We observe that GS and GAS require up to 3 hours; however, this is still within the same order (for more details, see Table 13b).

In Table 14 we also provide more details about training time of our methods for both pixel and latent space models.

## C.3 MEMORY USAGE

We are investigating the peak-memory GS/GAS required for training iteration depending on NFE.

In Table 15 we demonstrate the peak-memory usage for GS/GAS compared to LD3. When measuring the memory, we used the config we further report in Appendix D. GS requires the same amount of peak-memory allocated as LD3.

Incorporation of the discriminator loss into the training process of GAS only requires additional less than 4 gigabyte of memory usage, which is a minor overhead, especially considering its efficiency in terms of the final generation quality. This overhead is limited to training at inference time, GAS and GS sample at the same speed. Additionally, storing prior states does not incur additional overhead for peak-memory usage.

Table 13: Comparison of different training-based methods in terms of computational effectiveness across both pixel and latent space selected dataset.

(a) CIFAR10

| Method | NFE | FID | Time | GPU Type |
|---|---|---|---|---|
| CD | 2 | 2.93 | 8 days | A100 |
| PD | 8 | 2.47 | 8 days | TPU |
| S4S-Alt | 7 | 2.52 | < 1 hour | A100 |
| S4S | 10 | 2.18 | < 1 hour | A100 |
| LD3 | 10 | 2.32 | < 1 hour | A100 |
| GS | 6 | 2.44 | < 1 hour | H100 |
|  | 10 | 2.14 | < 1 hour | H100 |
| GAS | 4 | 3.98 | < 2 hours | H100 |

(b) Imagenet-256

| Method | NFE | FID | Time | GPU Type |
|---|---|---|---|---|
| LD3 | 4 | 9.19 | < 1 hour | A100 |
|  | 5 | 5.03 |  |  |
|  | 6 | 4.46 |  |  |
|  | 7 | 4.32 |  |  |
| GS | 4 | 7.97 | < 1.5 hours | H100 |
|  | 5 | 4.94 | < 2 hours |  |
|  | 6 | 4.29 | < 2 hours |  |
|  | 7 | 4.16 | < 2.5 hours |  |
| GAS | 4 | 6.06 | < 3 hours | H100 |

Table 14: Approximate training time (in minutes) for 10k iterations scenarios for GS and GAS in both pixel and latent space. For MS-COCO we use 1k iterations scenarios. All the numbers reported are computed using one H100 GPU.

(a) Pixel space models

|  |  | NFE=4 | NFE=6 | NFE=8 | NFE=10 |
|---|---|---|---|---|---|
| GS | CIFAR10 | 30m | 40m | 50m | 60m |
|  | FFHQ | 40m | 60m | 80m | 95m |
|  | AFHQv2 | 40m | 60m | 80m | 95m |
| GAS | CIFAR10 | 85m | 100m | 115m | 130m |
|  | FFHQ | 160m | 185m | 210m | 240m |
|  | AFHQv2 | 160m | 185m | 210m | 240m |

(b) Latent space models

|  |  | NFE=4 | NFE=5 | NFE=6 | NFE=7 |
|---|---|---|---|---|---|
| GS | LSUN | 35m | 45m | 50m | 60m |
|  | ImageNet | 75m | 95m | 115m | 135m |
|  | MS-COCO | 50m | 60m | 70m | 80m |
| GAS | LSUN | 125m | 140m | 150m | 165m |
|  | ImageNet | 185m | 210m | 245m | 270m |
|  | MS-COCO | 60m | 75m | 90m | 105m |

Table 15: Peak-memory usage (in gigabyte) for training iteration for GS and GAS in CIFAR10 and Imagenet-256. We use LD3 in our implementation. The official implementation uses LPIPS, rather than L1 distance in latent space as we do, which leads to the use of a VAE decoder at each step and incurs additional memory usage.

(a) CIFAR10

|  | NFE=4 | NFE=6 | NFE=8 | NFE=10 |
|---|---|---|---|---|
| GS | 17GB | 23GB | 28GB | 34GB |
| GAS | 19GB | 25GB | 30GB | 35GB |
| LD3 | 17GB | 23GB | 28GB | 34GB |

(b) Imagenet-256

|  | NFE=4 | NFE=5 | NFE=6 | NFE=7 |
|---|---|---|---|---|
| GS | 37GB | 45GB | 54GB | 62GB |
| GAS | 41GB | 49GB | 57GB | 66GB |
| LD3 | 37GB | 45GB | 54GB | 62GB |

## C.4 INFERENCE TIME

Inference process of our method requires additional operations performed with all prior states. However, they are incomparably computationally simpler than one step of diffusion model (function evaluation). Thus, the wall-clock time of inference for GS is comparable to the solver baselines, which we show in Table 16.

Table 16: Inference time in minutes for ImageNet dataset. We obtain the comparison by generating 1,024 images with batch 64 utilizing a single H100 GPU. GAS differs from GS only in the training process; their inference times are identical.

| Method | NFE=4 | NFE=5 | NFE=6 | NFE=7 |
|---|---|---|---|---|
| UniPC(3M) | 0.36m | 0.46m | 0.55m | 0.64m |
| GS (Ours) | 0.36m | 0.45m | 0.55m | 0.64m |

This pattern does not depend on the model and dataset choice; therefore, our method does not introduce any inference time overhead on both pixel, latent or text-to-image diffusion models.

## D EXPERIMENTAL DETAILS

### D.1 BASELINE DISCRETIZATION HEURISTICS

In this section, we provide the reader with the common timestep schedules, used in the paper.

**Polynomial discretization (time-quadratic, time-uniform)** defines the timestep schedule via a polynomial function of the uniform sequence. Specifically, it defines

$$t_i = \left(\frac{i}{N}\right)^\rho (T - t_{\text{eps}}) + t_{\text{eps}}, \quad i = 0, 1, \dots, N. \tag{23}$$

Here $\rho$ is often set to 1 or 2 (Song et al., 2020b; Ho et al., 2020; Song et al., 2020a) which corresponds to time quadratic and time uniform discretization.

**Time logSNR** schedule builds on top of the signal-to-noise ratio $\alpha_t^2/\sigma_t^2$. Specifically, log-SNR uses the transformation $\lambda_t = \log(\sigma_t/\alpha_t)$ and defines

$$\lambda(t_i) = \frac{N - i}{N}(\lambda_T - \lambda_{\text{eps}}) + \lambda_{\text{eps}}, \quad i = 0, 1, \dots, N. \tag{24}$$

This schedule offers high generation quality with different versions of the DPM-Solver (Lu et al., 2022a;b; Zheng et al., 2023).

**GITS schedule** provides an optimized sequence of noise levels for diffusion models, targeting very low NFE. Originally proposed in Chen et al. (2024a) for ODE-based diffusion processes with trajectory regularity constraints. We use optimized timesteps in Stable Diffusion experiments from Tong et al. (2024). Concretely, the timestep schedules are:

$$\text{NFE} = 4: \quad [1, 0.6837, 0.3673, 0.1176, 0.001];$$
$$\text{NFE} = 5: \quad [1, 0.7669, 0.4839, 0.2341, 0.0676, 0.001];$$
$$\text{NFE} = 6: \quad [1, 0.7836, 0.5504, 0.3340, 0.1508, 0.0343, 0.001];$$
$$\text{NFE} = 7: \quad [1, 0.8502, 0.6004, 0.4006, 0.2175, 0.0843, 0.0176, 0.001];$$
$$\text{NFE} = 8: \quad [1, 0.8502, 0.6504, 0.4672, 0.3007, 0.1675, 0.0676, 0.0176, 0.001].$$

### D.2 TEACHER SOLVER

**Data generation** For a fair comparison, we follow Tong et al. (2024) to generate the teacher dataset. We choose UniPC with the parameters used in LD3. We utilize class condition of the ImageNet-256 teacher and generate the corresponding dataset with the classifier-free guidance scale of 2.0 and generate 50 images per each of the 1000 classes. We report details in Table 17.

Table 17: Detailed description of the UniPC solver parameters used for a teacher dataset generation consisting of 50000 images for both pixel and latent space scenarios.

|  | CIFAR10 | FFHQ | AFHQv2 | LSUN-Bedroom-256 | Imagenet-256 |
|---|---|---|---|---|---|
| Order | 3 | 3 | 3 | 3 | 3 |
| NFE | 20 | 20 | 20 | 20 | 10 |
| Time schedule | logSNR | logSNR | logSNR | time-uniform | time-quadratic |
| $B(h)$ | bh1 | bh1 | bh1 | bh2 | bh2 |
| $t_{\text{eps}}$ | 1e-4 | 1e-4 | 1e-4 | 1e-3 | 1e-3 |
| FID | 2.03 | 2.60 | 2.16 | 3.06 | 4.10 |

**Stable Diffusion details**   Regarding text-to-image generation with Stable Diffusion, we observe that output image distributions of low-NFE students (NFE = 3-5) differ significantly from those of a high-NFE teacher (e.g., NFE = 10). Since such students have very few trainable parameters, direct distillation can be inefficient. The same pattern was found in Tong et al. (2024). For such reason and a fair comparison, we follow identical to the LD3 approach teacher generation protocol. We train student at NFE = $n$ with the teacher at NFE = $n + 1$. This "one-plus" teacher minimizes the gap in noise dynamics and yields smoother, more reliable convergence.

Moreover, in our experiments, we find that FID loses its correlation with perceived fidelity at high NFE, so we treat improvements in that regime with particular caution. Recognizing this unreliability beyond NFE $\approx 8$ reinforces our choice of simpler teachers as the most robust path to high-quality samples. Further details on teacher parameters are provided in Table 18.

Table 18: Detailed solver parameter settings for teacher-generated dataset using 30000 MS-COCO prompts.

| Student's NFE | NFE=4 | NFE=5 | NFE=6 | NFE=7 |
|---|---|---|---|---|
| Teacher's NFE | 5 | 6 | 7 | 8 |
| Solver | IPNDM(2M) | IPNDM(2M) | IPNDM(2M) | IPNDM(2M) |
| Time schedule | GITS | GITS | GITS | GITS |
| FID | 14.10 | 12.08 | 11.80 | 11.48 |

### D.3   SOLVER COEFFICIENTS PARAMETERIZATION

The detailed description of the **Generalized Solver** step is provided in Algorithm 1. Specifically, when all parameters $\phi$ are set to zero, the GS reduces exactly to DPM-Solver++(3M) (Lu et al., 2022b).

### D.4   PRACTICAL IMPLEMENTATION DETAILS

We define $W$, $H$, and $C$ as the width, height, and number of channels of an image, respectively. Similarly, $W'$, $H'$, and $C'$ represent the corresponding dimensions in the latent space for the Latent Diffusion model (Rombach et al., 2022).

**Optimizer and trainable parameters**   We update three primary parameter sets during training: $\theta$ defines the timestep schedule, $\phi$ defines the solver coefficients and $\xi$ acts as a correction to the timesteps that we evaluate the pre-trained model on. We use one optimizer for all parameter groups. We use time-uniform schedule for the initialization of parameters $\theta$. We initialize $\xi$ and $\hat{c}_{j,n}(\phi)$, $a_{j,n}(\phi)$ with zeros. We use the EMA version of the model parameters for evaluation and update the EMA weights after each training iteration.

**Evaluation**   We evaluate our models (Table 2a, 2b) using the FID score with 50 000 randomly generated samples. For ImageNet, we generate an equal number of samples for each class to ensure a balanced FID evaluation. We use EMA weights for evaluations. We calculate FID using reference

**Algorithm 1** Generalized solver (GS) with theoretical guidance from DPM-Solver++(3M). Denote $h_i = \lambda_{t_i^\theta} - \lambda_{t_{i-1}^\theta}$ for $i = 1, \ldots, N$.

1: $\psi_1 \leftarrow e^{-h_n} - 1$
2: $\mathbf{x}_{n+1} \leftarrow \left[a_{n,n}(t_{n:n+1}^\theta) + \hat{a}_{n,n}(\phi)\right] \cdot \mathbf{x}_n - \left[\alpha_{t_{n+1}}\phi_1 + \hat{c}_{n,n}(\phi)\right] \cdot \boldsymbol{v}(\mathbf{x}_n, t_n^\theta + \xi_n)$
3: **if** $n = 1$ **then**
4:     $r_0 \leftarrow \frac{h_{n-1}}{h_n}$
5:     $\mathbf{D1}_0 \leftarrow \frac{1}{r_0}\left[\boldsymbol{v}(\mathbf{x}_n, t_n^\theta + \xi_n) - \boldsymbol{v}(\mathbf{x}_{n-1}, t_{n-1}^\theta + \xi_{n-1})\right]$
6:     $\mathbf{x}_{n+1} \leftarrow \mathbf{x}_{n+1} - \left[\frac{\alpha_{t_{n+1}}\phi_1}{2} + \hat{c}_{n-1,n}(\phi)\right] \cdot \mathbf{D1}_0$
7: **else if** $n \geq 2$ **then**
8:     $r_0, \, r_1 \leftarrow \frac{h_{n-1}}{h_n}, \, \frac{h_{n-2}}{h_n}$
9:     $\psi_2 \leftarrow \frac{\psi_1}{h} + 1$
10:     $\psi_3 \leftarrow \frac{\psi_2}{h} - \frac{1}{2}$
11:     $\mathbf{D1}_0 \leftarrow \frac{1}{r_0}\left[\boldsymbol{v}(\mathbf{x}_n, t_n^\theta + \xi_n) - \boldsymbol{v}(\mathbf{x}_{n-1}, t_{n-1}^\theta + \xi_{n-1})\right]$
12:     $\mathbf{D1}_1 \leftarrow \frac{1}{r_1}\left[\boldsymbol{v}(\mathbf{x}_{n-1}, t_{n-1}^\theta + \xi_{n-1}) - \boldsymbol{v}(\mathbf{x}_{n-2}, t_{n-2}^\theta + \xi_{n-2})\right]$
13:     $\mathbf{D1} \leftarrow \mathbf{D1}_0 + \frac{r_0}{r_0+r_1}\left[\mathbf{D1}_0 - \mathbf{D1}_1\right]$
14:     $\mathbf{D2} \leftarrow \frac{1}{r_0+r_1}\left[\mathbf{D1}_0 - \mathbf{D1}_1\right]$
15:     $\mathbf{x}_{n+1} \leftarrow \mathbf{x}_{n+1} + \left[\alpha_{t_{n+1}}\phi_2 + \hat{c}_{n-1,n}(\phi)\right] \cdot \mathbf{D1} - \left[\alpha_{t_{n+1}}\phi_3 + \hat{c}_{n-2,n}(\phi)\right] \cdot \mathbf{D2}$
16: **end if**
17: $\mathbf{x}_{n+1} \leftarrow \mathbf{x}_{n+1} + \sum_{j=0}^{\max(n-1,0)} a_{j,n}(\phi) \cdot \mathbf{x}_j + \sum_{j=0}^{\max(n-3,0)} c_{j,n}(\phi) \cdot \boldsymbol{v}(\mathbf{x}_j, t_j^\theta + \xi_j)$

statistics and code from Karras et al. (2022). For MS-COCO (Table 2c) we obtain the FID score on 30 000 images using the same validation captions and FID reference statistics as in LD3 (Tong et al., 2024).

### D.4.1 PIXEL SPACE DIFFUSION ON CIFAR10, FFHQ, AND AFHQV2

- **Pre-trained diffusion model:**
    - EDM (Karras et al., 2022);
- **Teacher:**
    - UniPC solver, NFE = 20, logSNR schedule;
- **Discriminator R3GAN (Huang et al., 2024):**
    - Pre-trained CIFAR10 checkpoint for CIFAR10;
    - Pre-trained FFHQ-64 checkpoint for both FFHQ and AFHQv2;
    - Training in pixel space;
- **Image resolution:**
    - $W = H = 32$, $C = 3$ for CIFAR10;
    - $W = H = 64$, $C = 3$ for FFHQ and AFHQv2;
- **Training/validation dataset size:**
    - CIFAR10: 1400/1000 for GS and GAS;
    - FFHQ and AFHQv2: 1400/1000 for GS; 5000/1000 for GAS;
- **Solver training:**
    - $\mathcal{L}_{distill}$ is LPIPS;
    - $\mathcal{L}_{adv}$ with weight = 0.1 for CIFAR10 and weight = 1.0 for FFHQ and AFHQv2;
    - EMA decay = 0.999;
    - Batch size = 24;
    - Adam optimizer, lr = 0.001, betas = (0.9, 0.999), weight decay = 0.0;
    - Gradients are clipped by the norm of 1.0;

- **Discriminator training:**
  - Batch size $= 24$;
  - Adam optimizer, lr $= 0.00001$, betas $= (0.9, 0.999)$, weight decay $= 0.0$;
  - $\lambda_1$ and $\lambda_2$ in Equation 11 are equal to $0.1$;
- **Training duration:**
  - 10k iterations for GS/GAS;

### D.4.2   LATENT SPACE DIFFUSION ON LSUN-BEDROOM AND IMAGENET

- **Pre-trained diffusion model:**
  - LDM (Rombach et al., 2022);
- **Teacher:**
  - UniPC solver for both LSUN-Bedrooms and ImageNet;
  - NFE $= 20$ and time-uniform schedule for LSUN;
  - NFE $= 10$ and time-quadratic schedule for ImageNet;
- **Discriminator R3GAN (Huang et al., 2024):**
  - FFHQ-64 architecture with random initialization;
  - Training in latent space;
- **Image resolution:**
  - $W = H = 256$, $C = 3$;
  - $W' = H' = 64$, $C' = 3$;
- **Guidance scale:** $2.0$ (for ImageNet);
- **Training/validation dataset size:**
  - $1400/1000$ for GS;
  - $5000/1000$ for GAS;
- **Solver training:**
  - $\mathcal{L}_{distill}$ is L1 in latent space;
  - $\mathcal{L}_{adv}$ with weight $= 1.0$;
  - EMA decay $= 0.999$;
  - Batch size $= 8$;
  - Adam optimizer, lr $= 0.001$, betas $= (0.9, 0.999)$, weight decay $= 0.0$;
  - Gradients are clipped by the norm of $1.0$;
- **Discriminator training:**
  - Batch size $= 8$;
  - Adam optimizer, lr $= 0.00001$, betas $= (0.9, 0.999)$, weight decay $= 0.0$;
  - $\lambda_1$ and $\lambda_2$ in Equation 11 are equal to $0.1$;
- **Training duration:**
  - 30k iterations for GS/GAS;

### D.4.3   TEXT-TO-IMAGE GENERATION WITH STABLE DIFFUSION

- **Pre-trained diffusion model:**
  - Stable Diffusion v1.5 (Rombach et al., 2022);
  - Gradient checkpointing at every UNet inference;
- **Teacher:**
  - NFE $= n + 1$, where $n =$ student NFE;
  - IPNDM(2M) solver with GITS;
- **Discriminator R3GAN (Huang et al., 2024):**

- FFHQ-64 architecture with random initialization;
- First convolution layer modified to accept 4-channel latent inputs;
- Training in latent space;

- **Image resolution:**
    - $W \times H = 512 \times 512, C = 3$
    - $W' \times H' = 64 \times 64, C' = 4$
- **Guidance scale:** $7.5$;
- **Training/validation dataset size:**
    - 1400/128 for GS;
    - 5000/128 for GAS;
- **Solver training:**
    - $\mathcal{L}_{distill}$ is L1 in latent space;
    - $\mathcal{L}_{adv}$ with weight $= 1.0$;
    - EMA decay $= 0.999$;
    - Batch size $= 4$;
    - Adam optimizer, lr $= 0.001$, betas $= (0.9, 0.999)$, weight decay $= 0.0$;
    - Gradients are clipped by the norm of $1.0$;
- **Discriminator training:**
    - Batch size $= 4$;
    - Adam optimizer, lr $= 0.00001$, betas $= (0.9, 0.999)$, weight decay $= 0.0$;
    - $\lambda_1$ and $\lambda_2$ in Equation 11 are equal to $0.1$;
- **Training duration:**
    - 1k iterations for GS;
    - 2k iterations for GAS;

## E    TIMESTEPS VISUALIZATION

To validate the adequacy of the learned decoupled timesteps, we provide plots showing both $t^\theta$ and decoupled $t^\theta + \xi$ timestep schedules for GS and GAS across several NFE settings on FFHQ and ImageNet (Figure 7).

The decoupled timesteps should remain positive and monotonically decreasing, and we observe that they consistently satisfy these common-sense characteristics without any explicit constraints during training. They never become negative and always decrease monotonically; moreover, they are typically slightly lower than the corresponding unconstrained timesteps.

In one experiment, the first timestep was slightly above 1, but this caused no issues: the denoising model uses continuous positional embeddings and is robust to small shifts, so such a value is not out of distribution.

## F    ADDITIONAL SAMPLES

To further demonstrate the method's competitive results, we provide the reader with the additional samples of GS and GAS, compared to the teacher and the baseline UniPC with the same NFE. For all models/datasets except Stable Diffusion, we choose samples corresponding to 6 random seeds (marked as "random") and 6 samples that are the most distinguishable between GS and GAS in terms of pixel-space $L_1$ distance (marked as "selected"). We choose the selected sample seeds at NFE $= 4$ and report the corresponding samples for all NFE. We report the samples for FFHQ (Figures 8, 9, 10, 11), AFHQv2 (Figures 12, 13, 14, 15), LSUN Bedroom (Figures 16, 17, 18, 19) and ImageNet (Figures 20, 21, 22, 23).

Most random samples show only minor fine-grained differences between GS and GAS (which is still important and has a positive effect on FID, as indicated in Table 2). At the same time, the selected

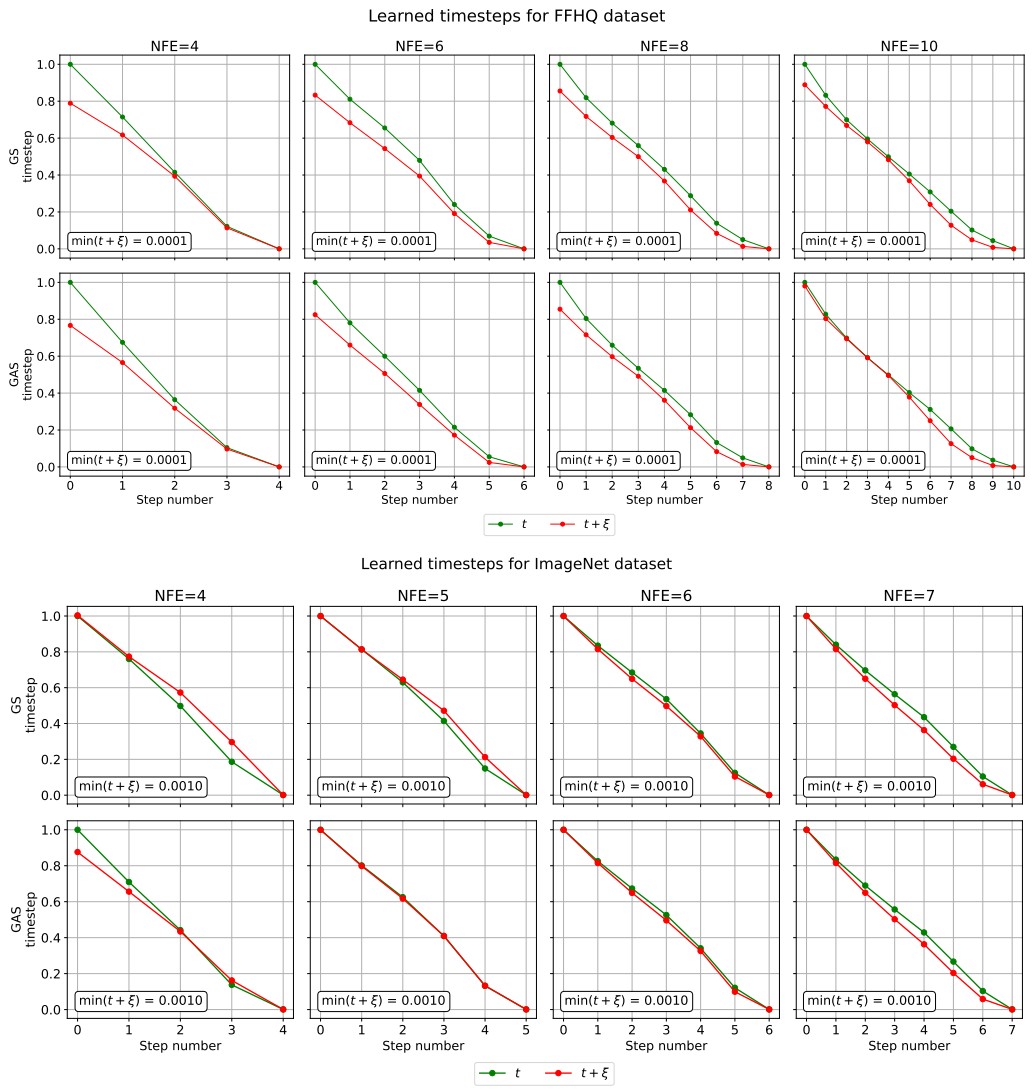

Figure 7: Visualization of final timestep schedules with and without additive corrections on FFHQ and ImageNet datasets. It is important to note that they never violate any of the common-sense characteristics expected of timestep schedules.

samples fully demonstrate the potential effect of the adversarial loss on the image quality. Most GAS samples at NFE = 4 demonstrate superior image quality compared to GS, while being farther from teacher. This further complements the results demonstrated in Figure 4. At the same time, one could tell that the pictures enhanced by adversarial loss differ depending on NFE: pictures from the same random seeds become significantly closer to the teacher starting from NFE = 6. This also indicates that the effect of the adversarial loss is the most prominent at low NFE, where it is harder for the student to replicate teacher's performance.

**Mode collapse** It is also worth noting that incorporation of the adversarial loss into the training process does not lead to mode collapse — a common concern in such cases — as we explicitly address this issue using the relativistic GAN loss from Huang et al. (2024). The random samples reported in Figures 8- 25 show generation diversity, while low resulting FID values indicate both high quality of our images and the absence of mode collapse.

**Stable Diffusion** For the Stable Diffusion experiments, we generate images from the 250 MS-COCO-val prompts with both the official LD3 implementation and our GAS method, initializing

both with identical random latent noise. From these outputs, we select six images at random (marked "random") and six that best highlight the visual differences between GAS and LD3 (marked "selected").

**Random prompts:**

- "A woman sitting on a bench and a woman standing waiting for the bus."
- "jumbo jet sits on the tarmac while another takes off"
- "An old green car parked on the side of the street."
- "A gas stove next to a stainless steel kitchen sink and countertop."
- "A person walking through the rain with an umbrella."

**Selected prompts:**

- "A man in a wheelchair and another sitting on a bench that is overlooking the water."
- "A fireplace with a fire built in it."
- "A half eaten dessert cake sitting on a cake plate."
- "an airport with one plane flying away and the other sitting on the runway"
- "A dirt bike rider doing a stunt jump in the air"

The resulting comparisons are shown in Figures 24, 25.

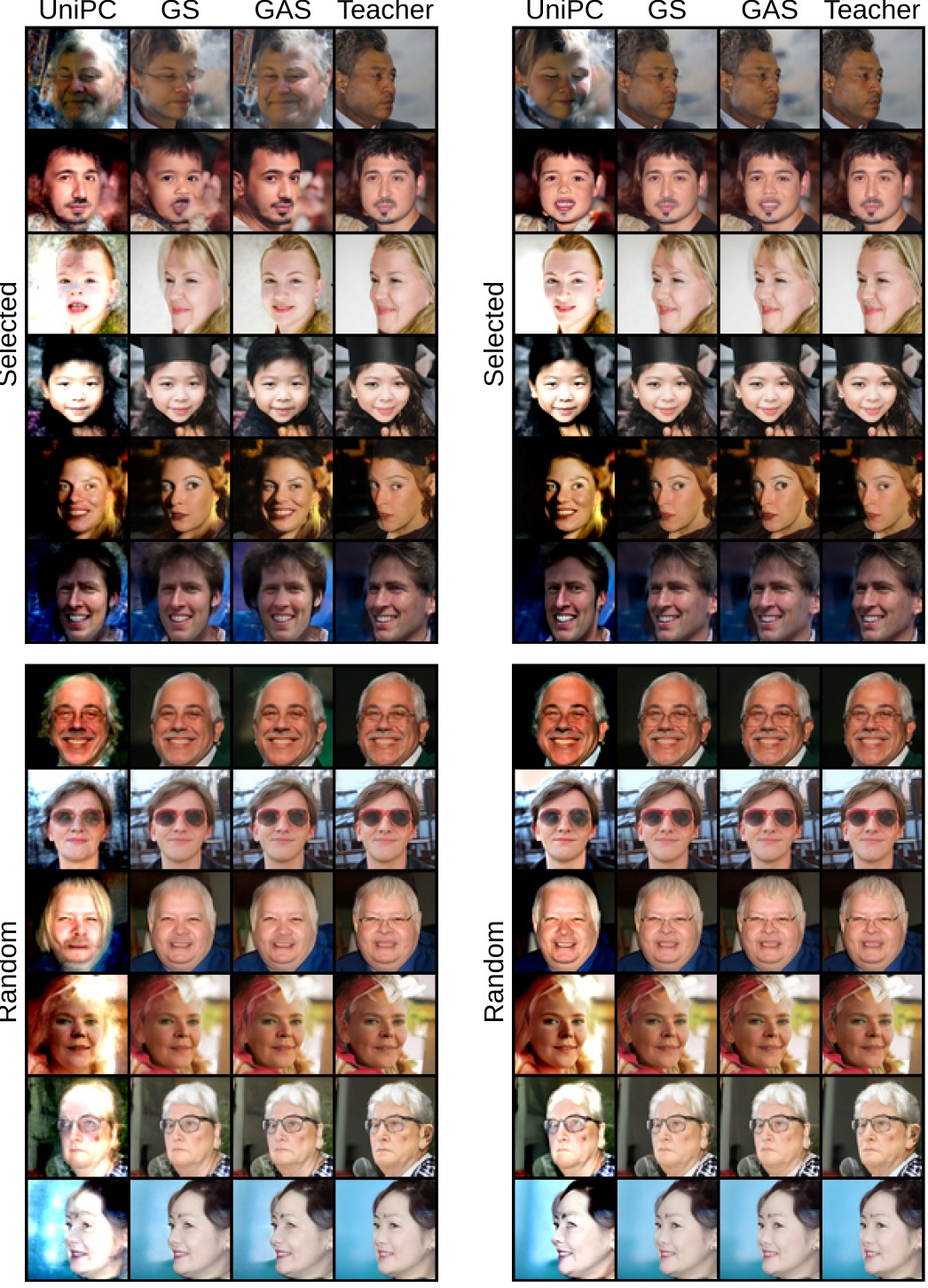

Figure 8: Comparison of GS and GAS with the teacher and UniPC on FFHQ with NFE = 4.

Figure 9: Comparison of GS and GAS with the teacher and UniPC on FFHQ with NFE = 6.

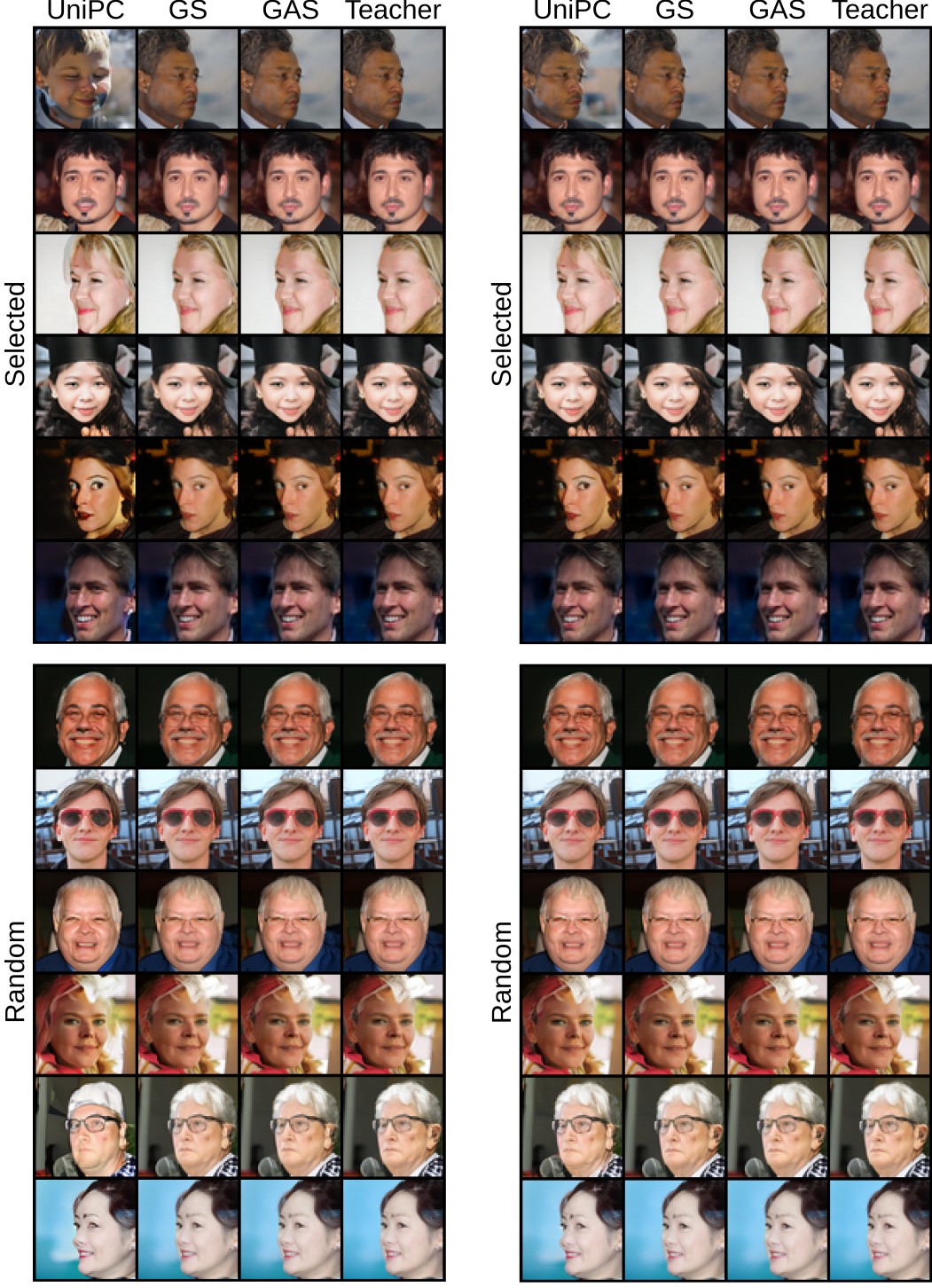

Figure 10: Comparison of GS and GAS with the teacher and UniPC on FFHQ with NFE = 8.

Figure 11: Comparison of GS and GAS with the teacher and UniPC on FFHQ with NFE = 10.

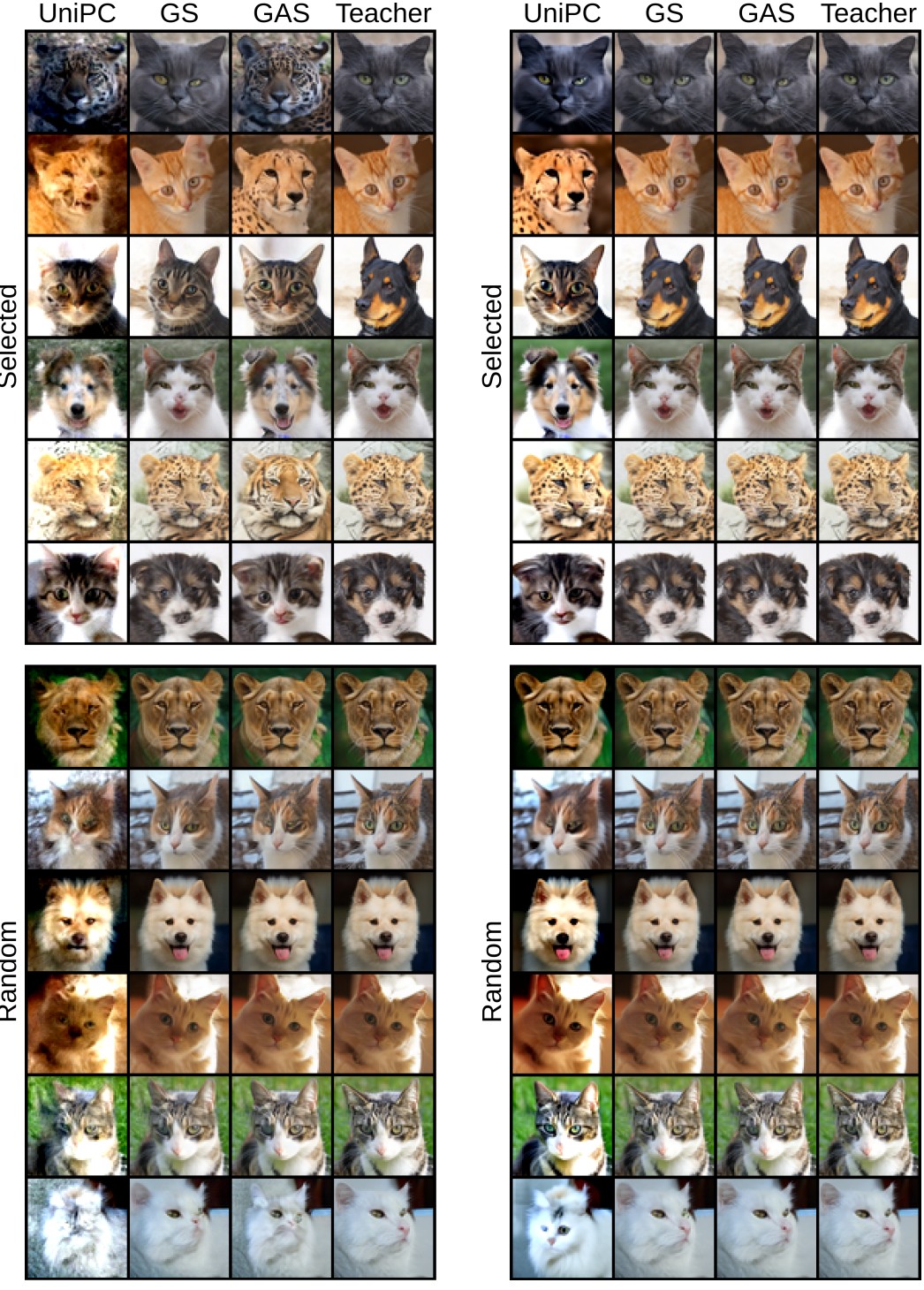

Figure 12: Comparison of GS and GAS with the teacher and UniPC on AFHQv2 with NFE = 4.

Figure 13: Comparison of GS and GAS with the teacher and UniPC on AFHQv2 with NFE = 6.

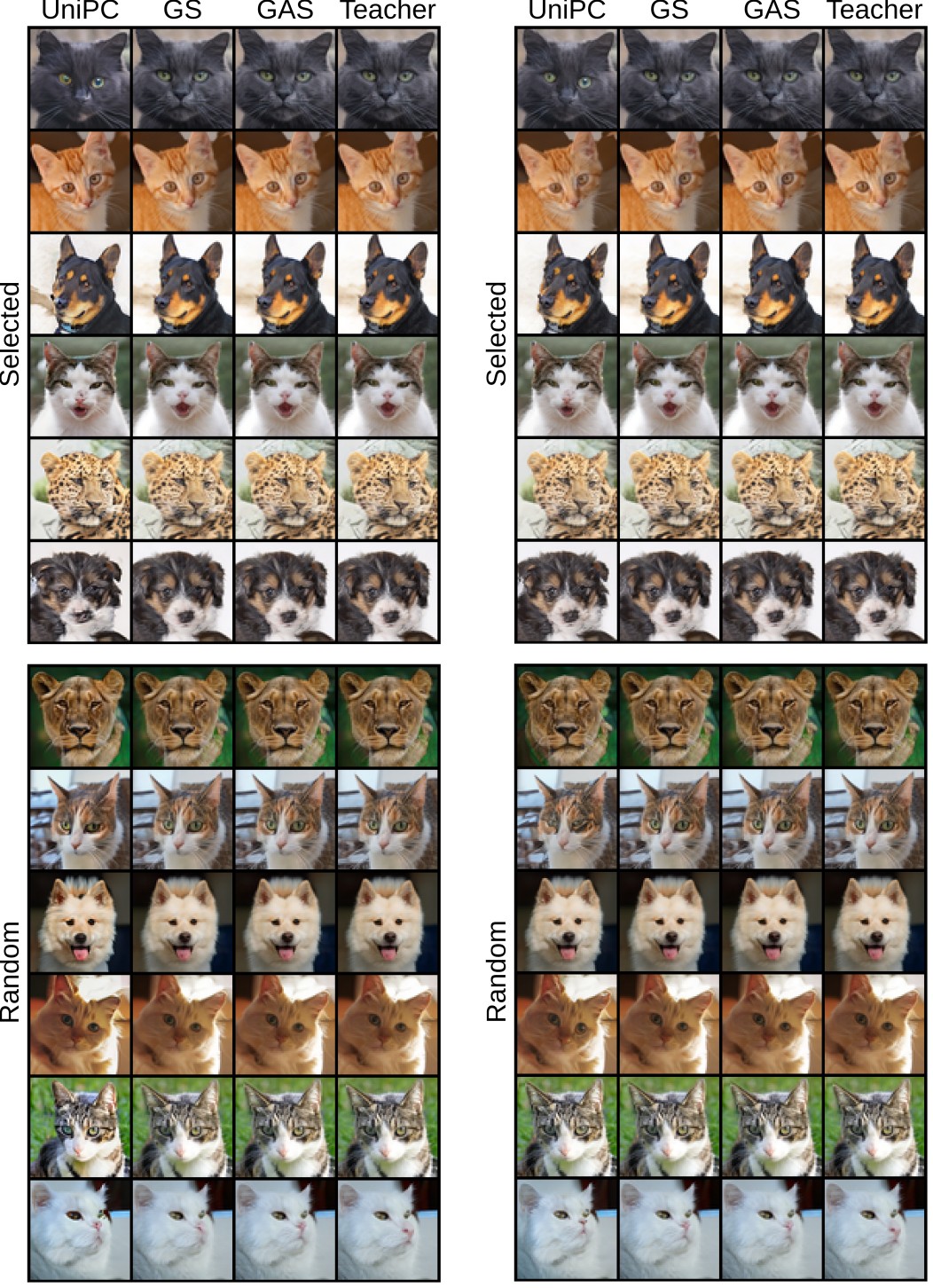

Figure 14: Comparison of GS and GAS with the teacher and UniPC on AFHQv2 with NFE = 8.

Figure 15: Comparison of GS and GAS with the teacher and UniPC on AFHQv2 with NFE = 10.

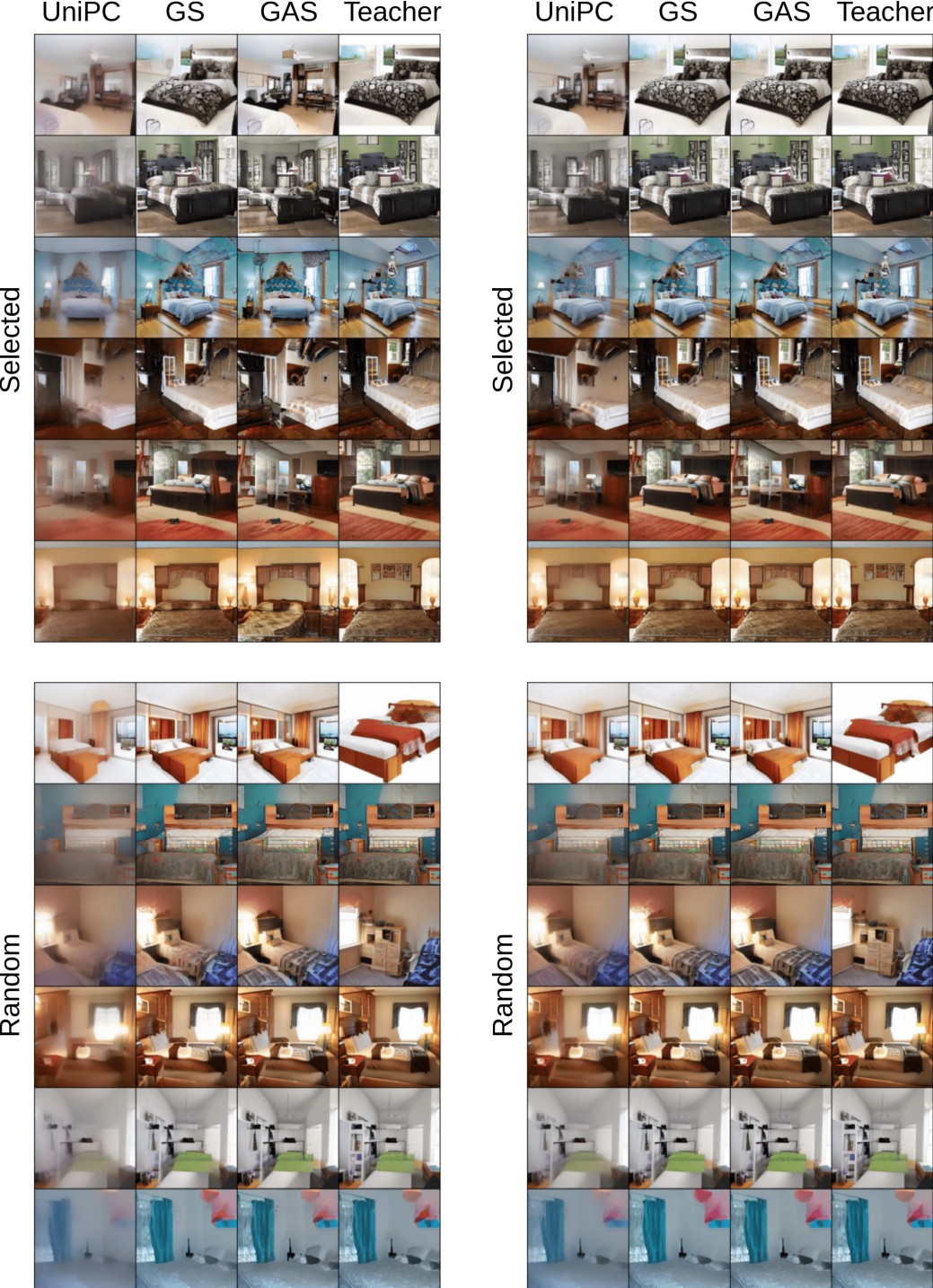

Figure 16: Comparison of GS and GAS with the teacher and UniPC on LSUN-Bedroom with NFE = 4.

Figure 17: Comparison of GS and GAS with the teacher and UniPC on LSUN-Bedroom with NFE = 5.

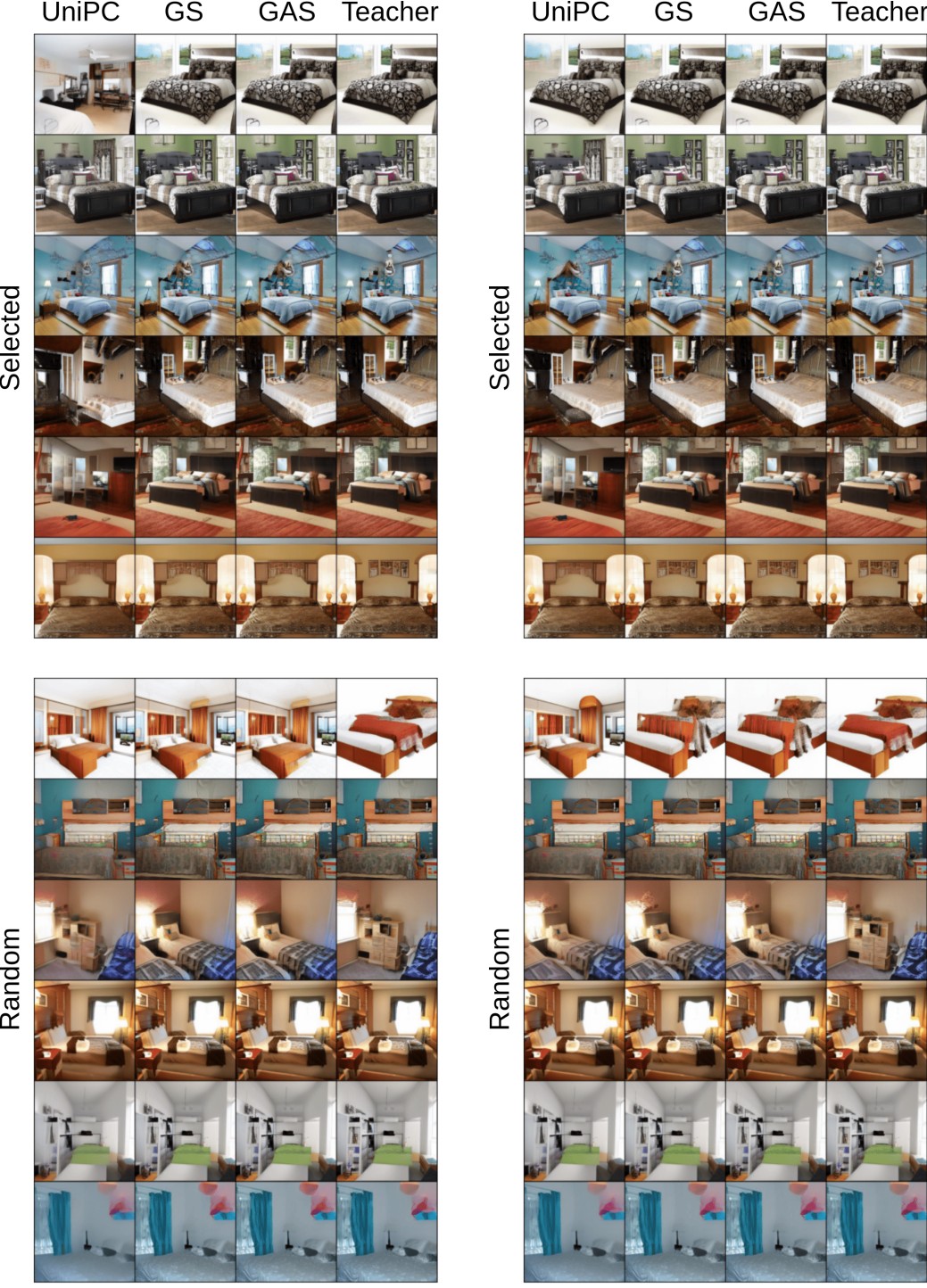

Figure 18: Comparison of GS and GAS with the teacher and UniPC on LSUN-Bedroom with NFE = 6.

Figure 19: Comparison of GS and GAS with the teacher and UniPC on LSUN-Bedroom with NFE = 7.

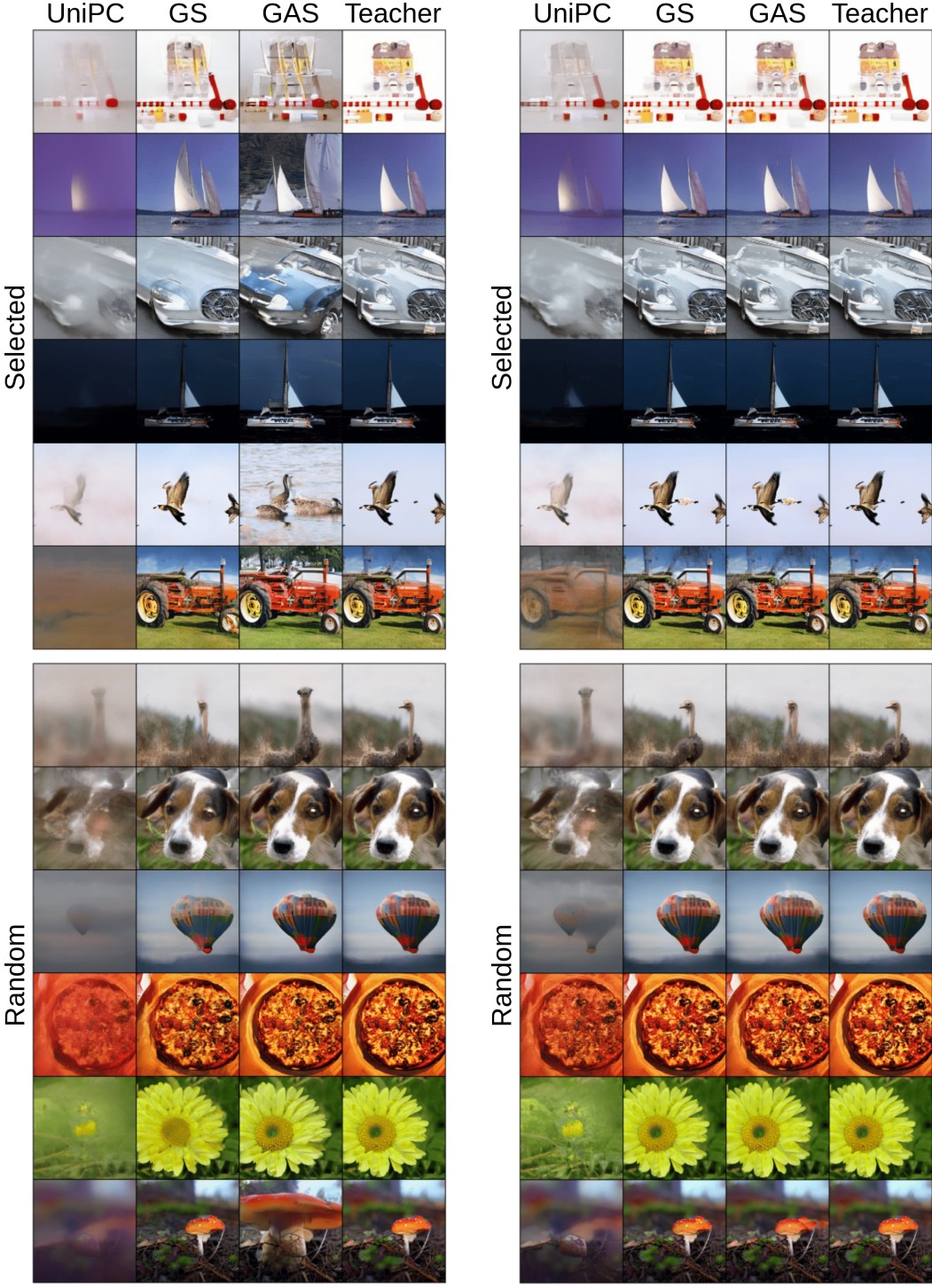

Figure 20: Comparison of GS and GAS with the teacher and UniPC on ImageNet with NFE = 4.

Figure 21: Comparison of GS and GAS with the teacher and UniPC on ImageNet with NFE = 5.

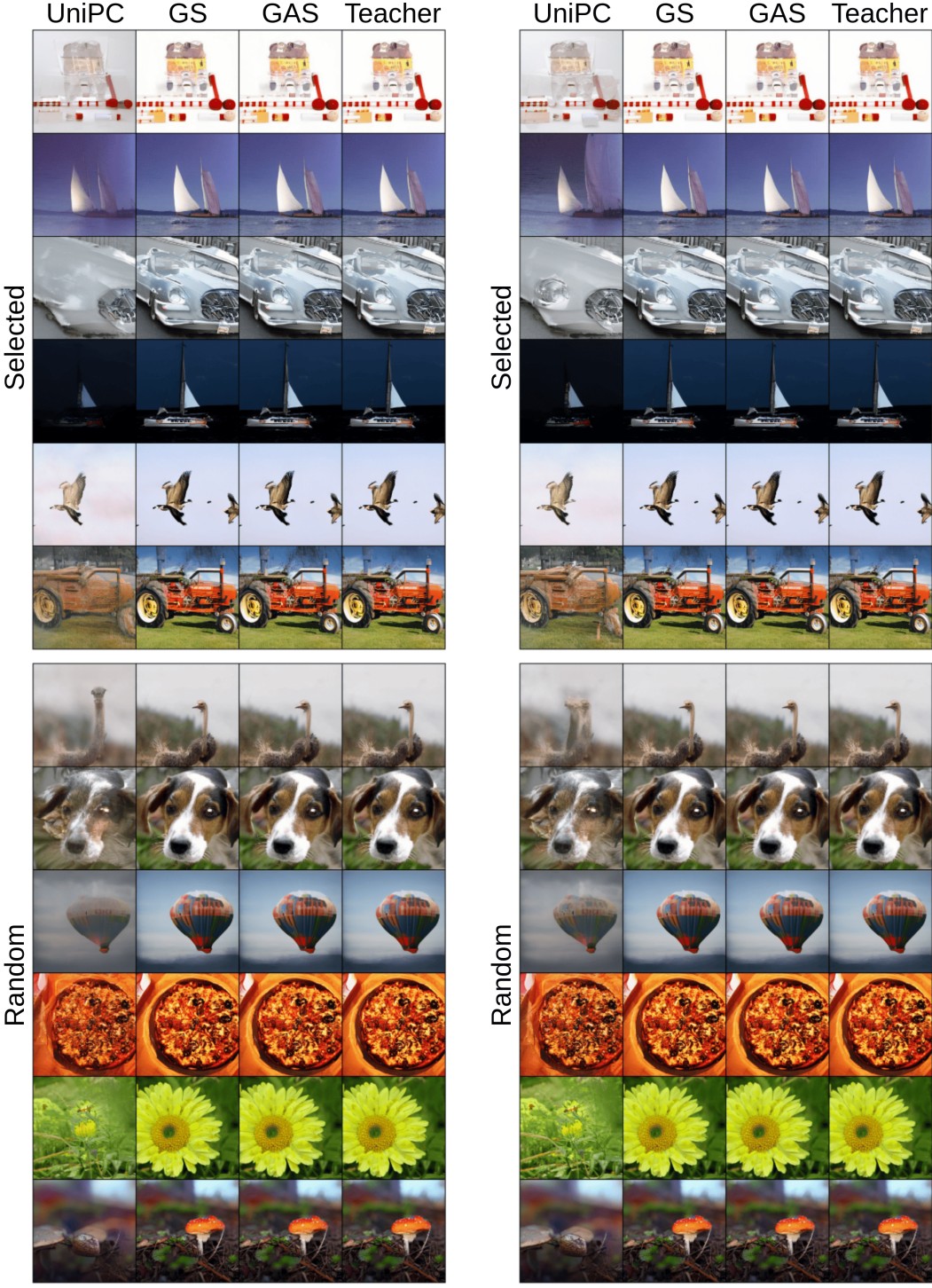

Figure 22: Comparison of GS and GAS with the teacher and UniPC on ImageNet with NFE = 6.

Figure 23: Comparison of GS and GAS with the teacher and UniPC on ImageNet with NFE = 7.

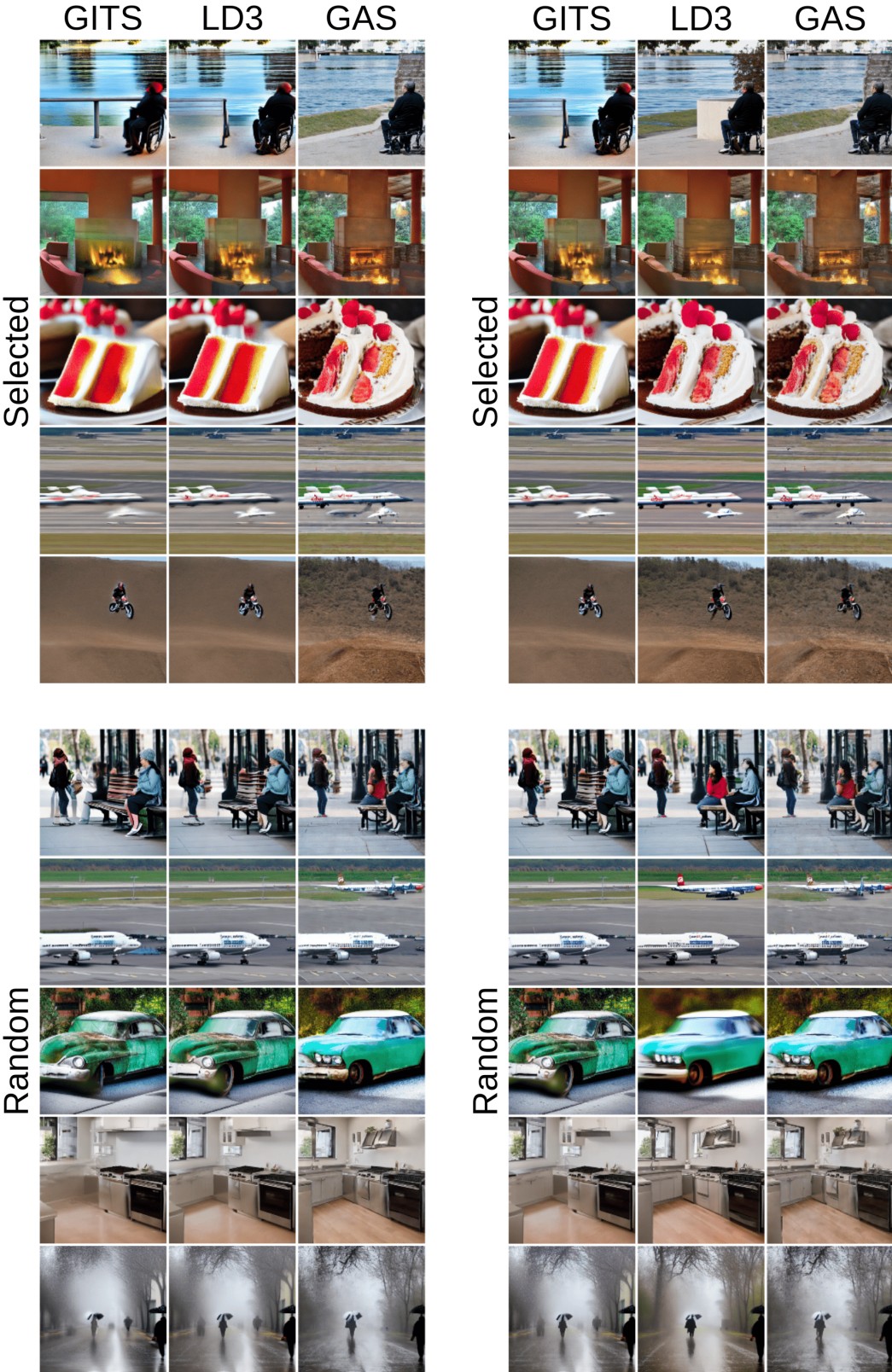

Figure 24: Comparison of GAS with LD3 and GITS on MS-COCO with NFE = 5.

Figure 25: Comparison of GAS with LD3 and GITS on MS-COCO with NFE = 6.

