# OpenReview forum: "GAS: Improving Discretization of Diffusion ODEs via Generalized Adversarial Solver"
_ICLR.cc/2026/Conference — ICLR 2026 Poster_

### Official Review · Reviewer_kJGM · 2025-10-28

**Soundness:** 3
**Presentation:** 3
**Contribution:** 2
**Rating:** 6
**Confidence:** 3

**Summary:**

The paper proposes an ODE solver distillation method for diffusion models that augments prior work with: an adversarial loss, a better initialization inspired by theory in other ODE solvers, and deliberate over parameterization. The goal is to improve sample quality at low NFE. Experiments and ablations indicate consistent gains over prior work, with the improvements most pronounced at low NFE. Ablations also isolate the utility of the adversarial loss and the theoretically sound initial parameterization. The work also provides useful evidence on cross dataset generalization and argues for faster training relative to consistency model or progressive distillation approaches.

**Strengths:**

- The work improves the performance of the ODE solver distillation paradigm.

- Thorough ablations demonstrate the contribution of each component, with the adversarial loss boosting performance at low NFE.

- Cross dataset generalization results in the appendix are strong and main text worthy.

- The appendix argument for faster training relative to consistency models or progressive distillation strengthens its practical appeal within the solver distillation family.

**Weaknesses:**

- The paper should explicitly state whether training is data free. By equations 7 and 22, no dependence on training data is apparent, yet in section 4.3 "Dataset size" is used and it is unclear. It seems to refer to the number of samples generated by the teacher, as parts of the appendix suggest.

- Line 475 claims the generalized solver parameterization "significantly accelerates training" relative to existing parameterizations. Section 4.2 compares final performance to S4S but does not show convergence speed. Appendix mentions the time required is similar to S4S (line 910).

- There exist methods that do not optimize the ODE solver but improve samples directly and can outperform solver distillation at very low NFE (for example Truncated Consistency Models [1]). The paper should acknowledge this limitation of the solver distillation class more clearly.

- The paper candidly acknowledges two important weaknesses shared by ODE solver distillation: per NFE training and backpropagation through the whole solver inference.

- Line 129 links "Table 2" to the table in this paper, but it seems to refer to a table in another work.

- The mode collapse discussion in the appendix line 1238 is hand wavy. Consider a quantitative method like coverage from Reliable Fidelity and Diversity Metrics for Generative Models [2]. (This is a minor issue as the claim is in the appendix and the method only changes the ODE-solver and not the main model)


[1] Lee, Sangyun, et al. "Truncated Consistency Models." The Thirteenth International Conference on Learning Representations.

[2] Naeem, Muhammad Ferjad, et al. "Reliable fidelity and diversity metrics for generative models." International conference on machine learning. PMLR, 2020.

**Questions:**

- Line 243: Please include the exact formula used for the initialization adapted to your notation. The cited paper does not provide sufficient detail on the higher order 3M variant.

- In Eq. 22, the meaning of p_T Y_T is not specified. Is it to indicate a different noise is sampled from the prior for the teacher.

- By Eqs. 7 and 22 it appears training does not depend on real training data. Then what does Sec. 4.3 "Dataset size" vary: number of noise samples, or a teacher generated dataset. Do you first generate a dataset with the teacher and then train on it? If so  are the generated samples used multiple times for training? (appendix suggests yes to both but it would be better to mention these details in main text as well)

- Did you test lower NFEs  than reported (one or two)? If so, what failed or succeeded?

- Do the learned time steps t_n + \xi_n increase with n? or does training learn a mixed or non monotone schedule?

---

> ### Author Response · Authors · 2025-11-23
> **Response to Reviewer kJGM - Part 1**
>
> Thank you for your detailed feedback. We appreciate the insights and address the identified weaknesses and questions below, providing clarifications and additional experiments where relevant.
>
> &nbsp;
>
> ### Weaknesses
>
> **__The paper should explicitly state whether training is data free. By equations 7 and 22, no dependence on training data is apparent, yet in section 4.3 "Dataset size" is used and it is unclear. It seems to refer to the number of samples generated by the teacher, as parts of the appendix suggest__**
>
> Yes, you are right; the term “dataset size” refers to the number of different samples generated by teacher model used in the student's training process.
>
> Our method is data-free, meaning that it does not require any dataset of real images (e.g., ImageNet). However, minimizing distillation loss is equal to minimizing a distance between teacher's and student's outputs generated from the same noise. The term “dataset size” refers to the number of pairs of initial noise inputs and the corresponding images generated by the teacher model.
>
> We appreciate your comment and will clarify this part in the camera-ready version of the paper.
>
> &nbsp;
>
> **_Line 475 claims the generalized solver parameterization "significantly accelerates training" relative to existing parameterizations. Section 4.2 compares final performance to S4S but does not show convergence speed. Appendix mentions the time required is similar to S4S (line 910)_**
>
> Thank you for your remark. Indeed, we made a misleading comment about Figure 3 in Section 4.2, which shows a comparison between our proposed parametrization and the one used in the S4S paper [1]. However, the figure is still informative and demonstrates that GS’s parametrization significantly outperforms the baseline in terms of validation distillation loss, which we emphasize in line 369. We will correct the wording.
>
> &nbsp;
>
> **_There exist methods that do not optimize the ODE solver but improve samples directly and can outperform solver distillation at very low NFE (for example, Truncated Consistency Models [2]). The paper should acknowledge this limitation of the solver distillation class more clearly_**
>
> Trainable ODE solvers require optimizing an impressively small number of parameters (fewer than 220 in all of our experiments) and can accelerate generation speed by several times, requiring only 1-5 hours of training on a single GPU.
>
> We agree that consistency models, such as the aforementioned Truncated CM [2], may perform better in very low-NFE setups. However, there is a fundamental methodological difference: our approach preserves the pre-trained weights of the diffusion model and trains only a tiny set of additional parameters, whereas consistency models contain millions of trainable parameters and require several GPU-days of training. Consequently, the two approaches operate under substantially different computational budgets and are not directly comparable.
>
> &nbsp;
>
>
> **_Line 129 links "Table 2" to the table in this paper, but it seems to refer to a table in another work_**
>
> In the raw LaTeX source of our paper, we use `~\citep[Table 2]{zhang2022fast}` to reference Table 2 in Zhang and Chen [3]. We attempted to reproduce the behaviour you described, and the same issue occurred when we opened the PDF with the Google Scholar browser extension. It automatically adds clickable links and, apparently, does so incorrectly in this case.
>
> &nbsp;
>
> **_The mode collapse discussion in the appendix line 1238 is hand wavy. Consider a quantitative method like coverage from Reliable Fidelity and Diversity Metrics for Generative Models [4]. (This is a minor issue as the claim is in the appendix and the method only changes the ODE-solver and not the main model)_**
>
> The incorporation of the adversarial loss does not lead to any mode-collapse issues. To validate this, we conducted the Reliable Fidelity and Diversity Metrics [4] evaluation for GS and GAS, comparing their statistics to those of the teacher model. We report results on the ImageNet dataset with NFE = 4.
>
> Table 1. Mode collapse ablation of the method
> |     | FID  | precision | recall | density | coverage |
> | --- | ---- | --------- | ------ | ------- |----------|
> | GS  | 7.87 | 0.90      | 0.63   | 1.18    | 0.90     |
> | GAS | 5.38 | 0.90      | 0.76   | 1.20    | 0.97     |
>
>
> We chose this setup because the incorporation of the adversarial loss was significant in this experiment and could raise the greatest concerns regarding mode-collapse. As shown in the table, GAS does not suffer from the aforementioned problem and even increases mode coverage and density compared to GS, which is trained without adversarial loss.
>
> Thank you for your remark. We will add this experiment to the camera-ready version of the paper to make the mode-collapse discussion clearer and more persuasive.

---

> ### Author Response · Authors · 2025-11-23
> **Response to Reviewer kJGM - Part 2**
>
> ### Questions
>
> **_Line 243: Please include the exact formula used for the initialization adapted to your notation. The cited paper does not provide sufficient detail on the higher order 3M variant_**
>
> The DPMSolver++(3M) initialization is taken from the official implementation on [GitHub](https://github.com/LuChengTHU/dpm-solver). To clarify the exact formulations, we have provided the pseudo-code for parameter initialization and all solver steps in the originally submitted version of the paper, which can be found in Appendix D.4, Algorithm 1.
>
> &nbsp;
>
> **_In Eq. 22, the meaning of $p_T(y_T)$ is not specified. Is it to indicate a different noise is sampled from the prior for the teacher_**
>
> Yes, you are correct. In Equation 22, $p_T(y_T)$ denotes the prior distribution (standard Gaussian). We use the notation $p_T(y_T)$ in addition to $p_T(x_T)$ to indicate that different initial noises are used for student and teacher generation in the discriminator loss calculation. This follows the R3GAN [5] training process.
>
> We will include this clarification to the camera-ready version of the paper.
>
> &nbsp;
>
> **_By Eqs. 7 and 22 it appears training does not depend on real training data. Then what does Sec. 4.3 "Dataset size" vary: number of noise samples, or a teacher generated dataset. Do you first generate a dataset with the teacher and then train on it? If so are the generated samples used multiple times for training? (appendix suggests yes to both but it would be better to mention these details in main text as well)_**
>
> Our method does not depend on real training data, as we have already mentioned in the Weaknesses. To obtain data for the student’s training, we sample a set of random initial noises and generate teacher images from them. The set of such noise-image pairs is treated as the dataset in our task; therefore, the dataset size refers to the number of these pairs.
>
> Datasets are pre-generated beforehand for training, since teacher generation is computationally demanding and should be done separately from the training process. The dataset is fixed to 1400 or 5000 samples, depending on the setup, while models are trained for several thousand iterations with a batch size of 4 or more. Thus, each dataset sample is used multiple times during training.
>
> &nbsp;
>
> **_Did you test lower NFEs than reported (one or two)? If so, what failed or succeeded?_**
>
> Yes, we conducted experiments with NFEs and corresponding solver orders equal to 1 and 2.
>
> Table 2. Experiments on low-NFE setups.
> |                 | NFE=1 | NFE=2 | NFE=4 |
> | ----------------| ----- | ----- | ----- |
> | DPM-Solver++(3M)| 314.95| 134.36| 46.14 |
> | GS              | 147.56| 55.97 | 10.70 |
> | GAS             | 193.46| 60.78 | 7.86  |
> | N parameters    |   3   | 11    | 39    |
>
> We trained both GS and GAS to convergence and compared them with DPM-Solver++(3M) using the same number of NFEs and solver orders. By ‘N parameters,’ we refer to the number of trainable parameters in the Generalized Solver for the corresponding NFE.
>
> As seen from the table, the method shows limited success in such low-NFE setups. Recreating a teacher’s trajectory with only one or two steps is a very challenging task for the solver. We believe this occurs because the number of trainable parameters is too limited in low-NFE cases, and the initial teacher model’s trajectory is not straight enough to be accurately replicated.
>
> It is also interesting to note that incorporating an adversarial loss in these setups does not improve generation quality. We assume this occurs because the student’s generated images are too noisy and contain many artifacts, causing the discriminator to collapse on them.
>
> &nbsp;
>
> **_Do the learned time steps $t_n + \xi_n$ increase with $n$? or does training learn a mixed or non monotone schedule?_**
>
> Thank you for your question. We now provide plots of the learned timesteps in the Section F of the Appendix, Figure 25. This makes the overall picture more complete.
>
> While investigating the final learned timesteps, we noted that the schedule always follows common sense. It never produces negative values, and the timesteps always decrease monotonically. The first timestep was once slightly greater than 1, but this did not cause any problems. The denoising model receives timesteps as positional embeddings, which are continuous and do not change dramatically if a timestep is slightly shifted. Therefore, this is not an out-of-distribution value for the denoising model.
>
> &nbsp;
>
> ## References
>
> [1] Eric Frankel, et al. "S4S: Solving for a Diffusion Model Solver."
>
> [2] Sangyun Lee, et al. "Truncated Consistency Models."
>
> [3] Qinsheng Zhang and Yongxin Chen. "Fast Sampling of Diffusion Models with Exponential Integrator."
>
> [4] Muhammad Ferjad Naeem, et al. "Reliable Fidelity and Diversity Metrics for Generative Models."
>
> [5] Yiwen Huang, et al. "The GAN is dead; long live the GAN! A Modern GAN Baseline".

---

> > ### Comment · Reviewer_kJGM · 2025-11-26
> >
> > I thank the authors for detailed response and the additional experiments.
> > Regarding authors' point that "the two approaches operate under substantially different computational budgets and are not directly comparable." I remain unconvinced that the methods cannot be meaningfully compared, for the following reasons:
> >
> > - A lower parameter count does not inherently imply a substantially lower training cost. GAS trains a discriminator and requires gradients to pass through the original model across steps (even if activations do not need to be stored). With PEFT methods such as LoRA the computational requirements of distillation can be comparable. For example, the difference between LCM-LoRA training time and GAS does not appear to be large (stated in the response to reviewer JFWr). As this reviewer stated one can even compare the performance on both methods under similar training costs.
> > - The savings generated from having NFE=2 at inference time can compensate for the one time additional cost for distillation.
> >
> > Overall, in my assessment the paper offers an incremental contribution in a specific setting and its relevance for practical applications may currently be limited. I am maintaining my original score.

---

> ### Author Response · Authors · 2025-12-03
> **Official Comment by Authors - Part 1/2**
>
> Since reviewers JFWr and kJGM raised closely related questions, we address them together with one consolidated response that is repeated for each reviewer.
>
> We thank the reviewer for the detailed feedback. We would like to emphasize a fundamental distinction between GAS and distillation-based approaches. While distillation creates a new model by changing the weights of the original one, GAS trains only the solver, leaving the weights of the diffusion model unchanged. This key property allows GAS to fully preserve the generative characteristics of the original model.
>
> ## Distillation-based method uses at least 60 million parameters, whereas GAS uses fewer than 220 parameters. What's the significance of this difference?
>
> Research on the properties of generative models is presented in [1], which shows that Consistency Models may lag behind classical diffusion models in terms of subjective quality [1, Table 1] and diversity [1, Table 13]. Our experiments confirm this: GAS achieves higher diversity than distillation methods. Although the problem formulations differ, to directly address the reviewer's comment, we compared GS and GAS with distillation-based methods given equal training time and observed that our method demonstrates superior quality. We will add new comparison results with Consistency Models in Table 7a.
>
> To evaluate diversity, we compare the coverage [2] and FID of our method GS and GAS against the official checkpoint of CD [5] on CIFAR-10. We assess generation quality against a teacher solver [Appendix D.2], which achieves FID=2.03 and coverage=0.971. Throughout, when working with CD, we used the official implementation [3] and ternary search for NFE=4 [5, Section 3]. Coverage was measured between 10,000 generated images and the CIFAR-10 test set.
>
> The results of the comparison of FID and coverage are presented in Table 4 and show that GS and GAS have higher coverage than CD. In addition, CD demonstrates low coverage compared to the teacher solver, which indicates a deterioration in the generative properties of the diffusion model. Although 60M parameters are indeed relatively small, changing them through distillation can change the behavior of the model, which, as we have shown, harms diversity. Our method, without changing these parameters, provides a more flexible and property-preserving acceleration method.
>
> Table 4. Comparison of generative properties of models, CIFAR-10.
>
> |                      | CD, NFE=1 | CD, NFE=4 | GS, NFE=4 | GAS, NFE=4 | Our teacher |
> |----------------------|-----------|-----------|-----------|------------|-------------|
> | Coverage             | 0.942     | 0.938     |  0.963    |  0.961     |  0.971      |
> | FID                  | 3.56      | 2.99      |  4.41     |  4.05      |  2.03       |

---

> ### Author Response · Authors · 2025-12-03
> **Official Comment by Authors - Part 2/2**
>
> ## If time was equated (e.g. by taking fewer training steps during distillation), how do you think they would compare?
>
> Table 4 above compares fully-trained models. To directly address the reviewer's question about training cost, we additionally conducted experiments with matched training time, shown in Table 5 below. GS and GAS demonstrate higher generation quality for FID and coverage with equal training time. We trained CD from scratch using the official implementation [3] for 10,000 iterations, which took about 4 hours on 2 A100 GPUs. It took 19 GB of video memory and 4 hours of training on a single A100 to train GAS at NFE=4. Table 5 demonstrates that with similar training time, GS and GAS have better generation quality than CD at NFE=1 and NFE=4. In addition, GAS does not impose an overhead in time on inference compared to solvers; we show this in Appendix C.2.
>
> Table 5. Comparison of CD with GS, GAS with similar training time, CIFAR-10.
>
> |              | CD, NFE=1   | CD, NFE=4   | GS, NFE=4   | GAS, NFE=4   |
> |--------------|-------------|-------------|-------------|--------------|
> | Coverage     |  0.613      |  0.891      |  0.963      |  0.961       |
> | FID          |  55.24      |  8.64       |  4.41       |  4.05        |
> | Training time| 8h          | 4h          | 1h          | 3h           |
>
> These matched-time results directly address the reviewer's question: when training budgets are equalized, GS and GAS substantially outperform CD in both FID and coverage. For example, GAS achieves FID of 4.05 in 3 hours at NFE=4, while CD requires 4 hours to reach FID of 8.64, indicating that our solver-training approach is more sample-efficient during training while better preserving generative properties.
>
> Overall, we address the reviewer's concerns in two complementary ways. Conceptually, GAS trains only the solver while keeping the diffusion model’s weights fixed, whereas distillation modifies tens of millions of model parameters and can degrade generative behavior. Empirically, Tables 4 and 5 show that GS and GAS maintain higher coverage than CD at convergence and, under similar training time, achieve substantially better FID and coverage. Taken together, these results support our claim that trained solvers preserve the original model’s generative properties while offering superior efficiency under comparable compute budgets.
>
> ## General research direction of learning a solver.
>
> In future work, we plan to explore solvers that are conditioned on a class, prompt, or $x_t$. This direction can significantly expand the capabilities of trained solvers and open up new ways to accelerate generation while maintaining the generative properties of the original model. A promising application of trained solvers is their joint use with distillation-based methods. A striking example is Multistep Consistency Models [4], which enable integration with solvers [4, Section 3.1]. Further research into trained solvers will allow deeper exploration of the potential of such hybrid approaches, increasing their effectiveness and flexibility.
>
>
> ### References
>
> [1] Exposing flaws of generative model evaluation metrics
> and their unfair treatment of diffusion models
>
> [2] Reliable Fidelity and Diversity Metrics for Generative Models
>
> [3] https://github.com/openai/consistency_models_cifar10
>
> [4] Multistep Consistency Models
>
> [5] Consistency Models

---

### Official Review · Reviewer_bLXP · 2025-10-30

**Soundness:** 3
**Presentation:** 3
**Contribution:** 3
**Rating:** 4
**Confidence:** 4

**Summary:**

This paper introduces a novel approach to training solvers for fast sampling by combining a new parameterization with an adversarial loss. The proposed method enhances the alignment between the generated and teacher distributions, outperforming traditional teacher-matching losses.

**Strengths:**

1. **Effective Use of Adversarial Loss**
   The incorporation of an adversarial loss is both intuitive and impactful, substantially improving distribution alignment between the generated samples and the teacher model.

2. **Clarity and Organization**
   The paper is clearly written and well-structured, making it easy to follow the methodology and results.

3. **Flexible and Innovative Parameterization**
   The proposed parameterization, distinct from that of S4S [1], introduces additional flexibility to the solver. This results in measurable improvements in both performance and generalization.

**Weaknesses:**

1. **High Training Cost from Adversarial Loss**
   While the adversarial loss improves overall quality, it significantly increases the training cost. As shown in Table 8, the training time is more than double that of GS (without adversarial loss), raising questions about computational efficiency.

2. **Questionable Initialization Strategy**
   The solver learns coefficients for a linear multistep method but initializes some parameters from the DPM solver [2], which belongs to the exponential integrator family. Since exponential integrators differ fundamentally from linear multistep methods, this initialization may lead to inconsistent behavior and a theoretically unsound starting point.

3. **Dependence on a Pretrained GAN**
   The method’s reliance on a pretrained GAN introduces unnecessary complexity and limits its generalizability. This dependency may hinder reproducibility and make it difficult to apply the method across different architectures.

4. **Unfair Comparison with S4S**
   The reported comparisons with S4S [1] may be misleading, as solver performance heavily depends on the teacher model used. Variations in teacher configurations can skew the results, weakening the validity of the performance claims.

**Questions:**

1. **Ensure Fairness in Training Time Comparisons (Table 7)**
   Training times should be reported using the same GPU model to ensure fairness, as differences in hardware can significantly affect latency measurements. Additionally, comparing results under the same number of function evaluations (NFE) would provide a more accurate measure of computational efficiency.

2. **Align Training Iterations Across Methods (Appendix D.4.1)**
   In Appendix D.4.1, GS and GAS appear to have been trained for different numbers of iterations (e.g., 1k for GS vs. 2k for GAS in the text-to-image experiment). This inconsistency complicates fair comparison. Both methods should be trained for the same number of iterations to enable a meaningful evaluation.

3. **Clarify the Choice of Initialization Solver**
   The paper initializes coefficients using the DPM solver [2], which belongs to the exponential integrator family which might affect the theoretical soundess of initializing some coefficients with it.

4. **Absence of Alternating Training Strategy and Its Rationale**
   Unlike S4S [1], the proposed method does not employ an alternating training strategy for the timestep scheduler and solver coefficients. It would be valuable for the authors to elaborate on why their approach doesn't require alternating strategy to stabilize training.

5. **Retrain S4S for Consistent Comparison**
   The S4S results presented are taken directly from the original paper, yet S4S performance is highly sensitive to the choice of teacher model. For example, in the LSUN-Bedroom 256 dataset, S4S underperforms relative to other solvers due to the poor teacher performance used in S4S in the original paper. Retraining S4S under the same setup and teacher model would ensure a fair and unbiased comparison with the proposed GAS method.

---

## References
[1] **S4S: Solving for a Diffusion Model Solver.**
[2] **DPM-Solver: A Fast ODE Solver for Diffusion Probabilistic Model Sampling in Around 10 Steps.**

---

> ### Author Response · Authors · 2025-11-23
> **Response to Reviewer bLXP**
>
> We appreciate the reviewer’s feedback and have addressed each point in detail.
>
> ## W1. High Training Cost from Adversarial Loss.
>
> GAS roughly doubles training time compared to GS, but the absolute cost remains modest: about 4.5 H100-hours on ImageNet (10k iterations) and under 2 hours on MS-COCO, versus the hundreds of GPU hours often required by diffusion distillation methods (e.g., Distribution Matching Distillation reports ~64 A100 days). The 2$\times$factor mainly comes from architecture: GS trains a tiny <220-parameter module on a frozen diffusion backbone, whereas GAS additionally trains a multi-million-parameter discriminator that is discarded after training, so at inference both methods use the same NFE and have identical speed and memory footprint.
>
> ## W2. Questionable Initialization Strategy
>
> Using DPM-Solver++(3M) as initialization is both theoretically sound and practically effective for our learned solver. Although DPM-Solver++ is derived via an exponential reformulation of the probability-flow ODE, the DPM-Solver++(3M) update is an Adams–Bashforth–style linear multistep combination of several past network evaluations. From this perspective, DPM-Solver++(3M) is a specialized linear multistep method for a transformed PF-ODE rather than a fundamentally different exponential integrator.
>
> ## W3. Dependence on a Pretrained GAN
>
> The reviewer’s concern does not apply to our main setting: for all latent-diffusion experiments (sections D.4.2 and D.4.3), the discriminator is initialized with random weights, so our method does not fundamentally rely on any pretrained GAN.
>
> ## W4. Unfair Comparison with S4S
>
> Solver performance can indeed depend on the choice of teacher, but our main conclusions do not rely on potentially mismatched S4S teachers. In our setup, the primary comparisons use LD3 teachers with official implementations and published FID scores, so all methods share exactly the same teacher (including hyperparameters), while S4S entries marked with $\dagger$ are included only as approximate references because S4S does not release code or fully specified teacher hyperparameters; all substantive performance claims in the paper come from the LD3-based experiments where the teacher is fully aligned across methods.
>
> ## Q1. Ensure Fairness in Training Time Comparisons (Table 7)
>
> Hardware differences do affect wall-clock training time, but in our setting, as discussed in our response to “W1. High Training Cost from Adversarial Loss”, they do not change the practical conclusions about computational cost: even on an A100, GS and GAS are only about 2$\times$slower than on an H100 and each configuration still completes within a few GPU-hours, far below the hundreds of A100 GPU-hours often reported for diffusion distillation or full fine-tuning.
>
> ## Q2. Align Training Iterations Across Methods (Appendix D.4.1)
>
> Appendix D.4.1 reports 1k iterations for GS and 2k for GAS in the text-to-image setting, but we ensure fairness by training each method to convergence rather than enforcing the same iteration count. When we extended GS to 2k iterations, its FIDs at 1k and 2k were effectively unchanged across all NFEs (see table below), indicating GS had already converged by 1k and that using 1k for GS vs. 2k for GAS yields a fair and meaningful comparison.
>
> || NFE=4 | NFE=5 | NFE=6 | NFE=7 |
> |--|--|--|--|--|
> | GS, 1k| 14.94 | 11.97 | 11.71 | 11.32 |
> | GS, 2k| 14.94 | 11.95 | 11.71 | 11.33 |
>
> ## Q3. Clarify the Choice of Initialization Solver
>
> Please see our responce to “W2. Questionable Initialization Strategy”.
>
> ## Q4. Absence of Alternating Training Strategy and Its Rationale
>
> S4S adopts an alternating optimization scheme for time steps and solver coefficients because the authors observed that jointly learning both makes the optimization landscape substantially more complex and increases the risk of overfitting [S4S, Section 4.2]. In our setting, however, we train GAS in a single-stage, end-to-end fashion and empirically observe stable optimization and no overfitting even for long runs, so a joint update is sufficient and keeps the method simpler without sacrificing robustness.
>
> ## Q5. Retrain S4S for Consistent Comparison
>
> As discussed in our response to “W4. Unfair Comparison with S4S,” our substantive claims are based on LD3-teacher experiments where all methods share the same officially implemented teacher and hyperparameters, and S4S numbers are used only as contextual references and clearly marked whenever the teacher cannot be matched. Retraining S4S under our exact setup would in principle be ideal, but is not realistically possible because the original work does not release code or sufficiently detailed teacher configurations, so under these constraints the fairest option is to report the original S4S results with explicit caveats and ground our main GAS comparisons in the fully aligned LD3-based experiments.
>
> ## References
> [1] Improved Distribution Matching Distillation for Fast Image Synthesis

---

> > ### Comment · Reviewer_bLXP · 2025-11-24
> >
> > I thank the authors for their reply. However, I still have one main concern. Since the primary difference between GAS and S4S, beyond the introduction of the GAN during training, lies in the parameterization, I believe that a fair and carefully controlled comparison between the two is essential for accepting this paper. As mentioned previously, even when using the same teacher model, the performance of the student solver depends heavily on the quality of the teacher trajectories being distilled. Because the experiments for GAS and S4S are not conducted under fully similar settings since the teacher trajectory for S4S is different and no results for S4S using the same trajectories as GAS were reported, it is difficult to clearly assess the true gains of GAS over S4S. Moreover, Table 3 currently lacks sufficient detail to judge whether the comparison is fair. For example, it is not clear whether S4S and GAS are initialized identically (i.e. with DPM-Solver++ (3M)); this is important, since initialization has a substantial impact on S4S performance. Additionally, it would also be helpful to include an ablation on the alternating training strategy used in S4S, to more convincingly demonstrate that the claimed removal of the need for alternating updates indeed follows from the new parameterization rather than other factors. For these reasons, I have decided to maintain my current score.

---

> > > ### Author Response · Authors · 2025-12-03
> > >
> > > We are grateful for the reviewer's follow-up questions, which address key aspects of our work. In response, we conducted additional experiments directly comparing GS and GAS with S4S under matched settings, i.e., using the same teacher trajectories, hyperparameters, and initialization (see Appendix D.4 for details), compensating for the fact that S4S does not provide an official implementation or complete optimization hyperparameters. Under these conditions, GS consistently outperforms the LMS and LMS+PC parameterizations proposed in S4S, and alternating training does not improve GS.
> > >
> > > ## Comparison GS and S4S parameterizations under fully similar settings and teacher trajectory.
> > >
> > > The S4S paper introduces two parameterizations, LMS and LMS+PC. In Table 3 of our paper, we compare GS with LMS+PC when all methods are trained to convergence. Here, we further show that GS outperforms both LMS and LMS+PC on FFHQ under identical training settings. We initialize LMS and GS so that the corresponding solvers are initially equivalent to DPM-Solver++(3M). Table 1 shows that for all NFE values, GS achieves better sample quality, which complements the comparison reported in Table 3 of the main paper.
> > >
> > > Table 1. Comparison of S4S parameterizations with GS on FFHQ.
> > >
> > > |         | NFE=4 | NFE=6|NFE=8 |NFE=10 |
> > > |---------|-------|------|------|-------|
> > > |S4S LMS      | 16.93 | 5.91 | 4.36 |4.08   |
> > > |S4S LMS + PC | 46.22 | 23.96| 7.95 |4.01   |
> > > |GS       | 10.70 | 4.49 | 2.96 |2.67   |
> > >
> > >
> > > ## Ablation on the alternating training strategy.
> > >
> > > A key difference between GS and S4S is that GS does not rely on alternating training. Because there is no official S4S implementation, we implemented alternating training ourselves following Algorithm 2 in the S4S paper. With our hyperparameters, each optimization subproblem required 1,000 iterations. Table 2 reports the effect of alternating training with 5 alternating steps (for a total of 10k total iterations, as in Appendix D.4). Alternating training substantially improves LMS+PC, but does not improve GS or LMS, and under our settings GS achieves the best performance without alternating training.
> > >
> > > Table 2. Effect of alternating training on S4S and GS on FFHQ.
> > >
> > > |              | NFE=4 | NFE=6|
> > > |--------------|-------|------|
> > > |S4S LMS          | 16.93 | 5.91 |
> > > |S4S LMS, Alt     | 26.64 | 8.9  |
> > > |S4S LMS + PC     | 46.22 | 23.96|
> > > |S4S LMS + PC, Alt| 37.89 | 12.14|
> > > |GS                     | 10.70 | 4.49 |
> > > |GS Alt                 | 16.5  | 5.26 |
> > >
> > > We further examine the sensitivity of alternating training to the number of alternating steps. Table 3 reports results for different values of K, showing that performance does not improve monotonically with more alternating steps (e.g., LMS at K=8 vs. K=10). In contrast, optimizing GS leads to monotonic quality improvements (Figure 5(a), Appendix B.2). While more careful tuning might further improve S4S, the original paper does not specify training hyperparameters, including optimizer settings for alternating training. Thus, GS not only outperforms S4S under a comparable baseline configuration, but also avoids the need for a complex and potentially unstable alternating training strategy, which is an important practical advantage.
> > >
> > > Table 3. Dependence on the number of alternating steps on FFHQ.
> > >
> > > |             |      |NFE=4  |       |       |NFE=6 |     |
> > > |-------------|------|-------|-------|-------|------|-----|
> > > | K           | 5    |  8    |  10   | 5     | 8    | 10  |
> > > |S4S LMS Alt     | 26.64| 26.17 | 26.60 | 8.90  | 7.46 |6.47 |
> > > |S4S LMS + PC Alt| 37.89| 37.25 | 37.08 | 12.14 | 10.71|10.32|
> > > |GS Alt       | 16.50| 15.63 | 15.18 | 5.26  | 4.91 |5.01 |
> > >
> > >
> > > ## Controlled comparison between GS and S4S on ImageNet.
> > >
> > > To provide a more complete comparison with S4S, we also evaluated S4S, GS, and GAS on ImageNet under the same initialization and teacher trajectories. As shown in Table 4, GS and GAS achieve strong performance relative to S4S on ImageNet as well.
> > >
> > > Table 4. Comparison of S4S with GS and GAS under fully comparable settings and initialization on ImageNet.
> > >
> > > |        | NFE=4 | NFE=5 | NFE=6 | NFE=7|
> > > |--------|-------|-------|-------|------|
> > > |S4S LMS     | 6.58  | 5.09  | 4.68  | 4.24 |
> > > |S4S LMS Alt | 9.33  | 7.43  | 5.13  | 4.20 |
> > > |GS      | 7.87  | 4.93  | 4.30  | 4.17 |
> > > |GAS     | 5.38  | 4.87  | 4.32  | 4.17 |
> > >
> > > In summary, our controlled experiments confirm that the GS parameterization consistently surpasses S4S under matched settings across multiple datasets (FFHQ, ImageNet) and NFEs, achieving better sample quality. GS also achieves superior performance without relying on the unstable alternating training required by S4S, establishing GS as a more robust and efficient method.

---

### Official Review · Reviewer_JFWr · 2025-11-01

**Soundness:** 4
**Presentation:** 2
**Contribution:** 2
**Rating:** 2
**Confidence:** 3

**Summary:**

This work studies the topic of distilling diffusion models into a few-step solver. This line of work focuses on the compute-efficient setting in which the parameters of the *solver*, not the score function/network itself, are learned.

It proposes a new efficient distilled solver, the generalized adversarial solver (GAS), which puts up state-of-the art results in the 4-10 NFE range on several datasets.

GAS is a combination of two techniques
- The generalized solver (GS): by parameterizing the next step $x_{n+1}$ as a linear combination of (1) the entire history of step values $\sum_j^n a_{j, n} x_j$ and (2) the entire history of vector field evaluations $\sum_j^n c_{j, n}v(x_j, t_j)$. The coefficients $\{a_{j, n}, c_{j, n}\}$  are parameterized as additive corrections to theoretically derived coefficients from past work. This contrasts to previous work, which learns only coefficients for the latter.
- GAS is formulated by adding a relativistic GAN loss term to the standard distance loss (either LPIPS or L1) used to train GS.

**Strengths:**

I think this project has the ingredients to ultimately build a strong paper.
- The research direction seems logical to me - training the solver might be more efficient than training the model itself.
- Its two components, GS and the GAN loss, each provide an FID boost over past methods, with the GAN loss's boost being particularly impressive in the 4-6 NFE range.
	- The GAN loss in particular is a very natural import from other subfields of diffusion modelling.

**Weaknesses:**

However, to me the paper reads more like a technical report than a top conference paper. A couple of ideas are proposed and results are shown, but not much beyond that in terms of motivation or generalizable insight and analysis.
-  Very little motivation is given for the way GS is designed. The motivation given (I believe L225 -> 229) is not honestly not very clear. Are you just looking for ways to add more parameters to the solver or is there something more to it?
	- As someone familiar with diffusion but without specific expertise in distilling diffusion solvers, I found the technical part, Section 3.1 tough to read, in large part because there is not much setup and motivation for the exposited ideas.
	- (However, I do think the motivation for the adversarial loss is clearer, as it is simpler and more obvious.)
- Additionally, neither of the two components of the method are really probed or ablated to understand why they work so well.
	- For example, the decision to use additive vs. non-additive coefficients in GS could be ablated easily, but it is not. As could the decision to include the past history of $x_j$s (this is similar to but not the same as S4S). With little motivation and not much ablation, it's hard to glean generalizable insights from the work.
	- Additionally, the decision to use the relativistic GANs is not ablated over popular alternative GAN losses.
- No code has been provided.

In summary, the two individual components of the method - the parameterization of the solver and the adversarial loss - are incremental, and little motivation, analysis, or generalizable insights are provided. Although I believe someone will find it useful, I do not believe it meets the bar for ICLR.

**Questions:**

- In Table 2, you tagged S4S Alt as unfair due to a different teacher FID score. Where did you find the teacher FIDs for S4S? I did not see them in the S4S paper (but may be missing them).
 - Can you explain the advantage of this approach over PEFT-based distillation of a diffusion model? I would think PEFT was more tractable because there's no need to backpropagate through entire 4-10 step solves?

---

> ### Author Response · Authors · 2025-11-23
> **Response to Reviewer JFWr - Part 1**
>
> Thank you for your valuable feedback and for pointing out the areas of improvement in our manuscript.
>
> **_Very little motivation is given for the way GS is designed. The motivation given (I believe L225 -> 229) is not honestly not very clear. Are you just looking for ways to add more parameters to the solver or is there something more to it? For example, the decision to use additive vs. non-additive coefficients in GS could be ablated easily, but it is not. As could the decision to include the past history of $x_j$s (this is similar to but not the same as S4S). With little motivation and not much ablation, it's hard to glean generalizable insights from the work._**
>
> We thank the reviewer for his constructive criticism. We are sorry that the motivation for GS was not clear from the text. We will summarize the motivation for GS at the beginning of Section 3.1. For clarity, in this response we will present the motivation for each part of GS in more detail. Let us briefly recall the main design choices in GS:
> - the use of theoretical coefficients, which form the basis of GS and improve convergence;
> - the signature of the linear multistep method, on which many theoretical solvers are based, determines the use of the past history of $x_j$s;
> - additive parameterization, which connects the theoretical and trainable solver coefficients within the signature.
>
> As mentioned above, theoretical coefficients form the basis of GS and ensure stable learning and excellent quality. The theoretical coefficients establish a strong inductive bias, as they provide GS with knowledge of the theoretically correct coefficients (L241->244, L291->294). Thus, in the optimization process, the trained coefficients do not have to learn complex theoretical dependencies, since they are already embedded in the theoretical coefficients.
>
> We additionally conducted an experiment to visually demonstrate the effect of theoretical coefficients. We trained GS with and without theoretical coefficients on the FFHQ dataset. We initialized the parameters such that, before training, the solver in both cases was equivalent to DPM-Solver++(3M). Table 1 shows that the use of theoretical coefficients significantly improves the final quality.
>
> Table 1. The effect of theoretical coefficients on FFHQ, FID.
>
> |            | NFE=4 | NFE=6 | NFE=8 | NFE=10 |
> |------------|-------|-------|-------|--------|
> | w/  theory | 10.70 | 4.49  | 2.96  | 2.67   |
> | w/o theory | 15.35 | 10.25 | 5.54  | 4.52   |
>
>
> To be able to use theoretical guidance from modern theoretical solvers, the GS signature must generalize them. The signature defines the solver's step formula and the number of trainable parameters. We chose the linear multistep method [1], [Equation 13], as it is well-researched in the field of numerical methods and many well-known ODE solvers are special cases of it, for example, Adams-Bashforth [Equation 12]. Thus, the GS signature allows us to embed coefficients from a large set of different theoretical solvers. You rightly drew attention to the use of the history of $x_j$s. This decision is not arbitrary but follows directly from the choice of the linear multistep method signature. For more detailed discussion about the Linear multistep method, see L128->131.
>
> The additive parameterization is responsible for the relationship between the theoretical and trainable coefficients within the signature. The parameterization is c = $c_{\text{theory}}$ + $\Delta$ c, where $c_{\text{theory}}$ is the theoretical coefficient that ensures stability and serves as the basis, and $\Delta$ c is a trainable correction that allows the solver to adapt. This choice is mainly for convenience and clarity: it lets us view GS as learning residual corrections on top of a stable, theory-based solver, in the same spirit as residual connections or LoRA in deep learning.
>
> Other parametrizations (e.g., multiplicative) are also possible, and for completeness we report a small comparison in table below, which shows behavior very similar to the additive case. This suggests that performance is largely insensitive to the exact parametrization, so we do not view the additive form as a central design choice that warrants an extensive ablation study
>
> Table 2. The effect of parameterization on FFHQ, FID.
>
> |                | NFE=4 | NFE=6 | NFE=8 | NFE=10 |
> |----------------|-------|-------|-------|--------|
> | additive       | 10.70 | 4.49  | 2.96  | 2.67   |
> | multiplicative | 10.73 | 4.59  | 3.02  | 2.65   |
>
> The results show that the parameterization of coefficients has less of an effect on the efficiency of GS than the theoretical coefficients do.

---

> ### Author Response · Authors · 2025-11-23
> **Response to Reviewer JFWr - Part 2**
>
> **_Additionally, the decision to use the relativistic GANs is not ablated over popular alternative GAN losses._**
>
> We chose R3GAN [2] because its authors demonstrate state-of-the-art results without additional training techniques and provide a theoretical basis. Furthermore, the relativistic loss demonstrates better convergence [R3GAN, Figure 1] and improved mode coverage [R3GAN, Table 1] compared to the traditional loss.
>
> To answer your question directly, we conducted an ablation study on the ImageNet dataset at NFE=4, comparing traditional and relativistic losses. We used random initialization for the discriminator.Table 3 shows that GAS improves on the baseline GS with either loss, and that the relativistic loss enables significant improvements over the traditional loss.
>
> Table 3. Ablation study of GAN losses on ImageNet, FID.
>
> | GS  | GAS (traditional) | GAS (relativistic) |
> |-----|---------------|-------------|
> | 7.87|  6.33         | 5.38        |
>
> Employing a GAN loss enhances the final output quality. GAS demonstrates strong performance with both traditional and relativistic losses. Although the relativistic loss is optional, it leads to a superior model.
>
> &nbsp;
>
>
> **_In Table 2, you tagged S4S Alt as unfair due to a different teacher FID score. Where did you find the teacher FIDs for S4S? I did not see them in the S4S paper (but may be missing them)._**
>
> You are correct that S4S does not provide FID values for its teachers. We concluded that the teachers are different, as S4S used different hyperparameters than we did. We will correct the phrase "different FID score" to "difference in teacher hyperparameters." Thank you for pointing this out. We relied on the teachers from LD3 because published FID values are available for them, and an official implementation exists that reproduces the FIDs reported in the LD3 paper. In contrast, S4S does not publish its FID scores and lacks an official implementation, making it impossible for us to reproduce their teacher models. The differences are in the teacher hyperparameters:
>
> - **ImageNet:** LD3 uses a time-quadratic schedule [LD3, Appendix D.1.2], while S4S uses a time-uniform schedule [S4S, Appendix G.2];
> - **MS-COCO:** LD3 uses teachers with NFE + 1 [LD3, Appendix D.1.3], while S4S uses NFE = 10 [S4S, Appendix G.2];
> - **LSUN:** S4S noted that it was impossible to reproduce the LD3 teacher [S4S, Table 15 caption], but we did not encounter this problem.
>
> &nbsp;
>
>
> **_Can you explain the advantage of this approach over PEFT-based distillation of a diffusion model? I would think PEFT was more tractable because there's no need to backpropagate through entire 4-10 step solves?_**
>
> The main advantages of our method over PEFT-based distillation are threefold: a reduced number of trainable parameters, generalization across datasets, and faster convergence. These benefits derive from a fundamental methodological difference: our approach preserves the pre-trained weights of the diffusion model, whereas PEFT-based distillation necessitates their modification. Each of these points is elaborated below.
>
> For comparison, we use the experimental configuration reported in LCM-LoRA [3], a PEFT-based distillation approach. LCM-LoRA uses at least 60 million parameters [3, Table 1], whereas we train fewer than 220 parameters. While the optimization procedure does not prohibit additional LoRA training for the diffusion model, our goal is to solve the problem without altering the model's weights.
>
> In Appendix B.2, we demonstrate that GS and GAS transfer well between different diffusion models, whereas PEFT-based distillation requires retraining for each diffusion model. We believe that the ability to generalize across datasets serves as both an observation and a motivation for researching methods that do not alter the weights of the diffusion model. The study of GS and GAS transfer between diffusion models is a topic for future research and is of practical significance.
>
> In addition, our method is less demanding in terms of training time and the number of samples required. Our training run requires 1-10 hours on an A100, whereas the fastest training run for LCM-LoRA requires approximately 32 hours on an A100.
>
> &nbsp;
>
> **_No code has been provided._**
>
> We have added the code to the new revision.
>
> ## References
>
> [1] https://en.wikipedia.org/wiki/Linear_multistep_method#Definitions.
>
> [2] The GAN is dead; long live the GAN! A Modern GAN Baseline.
>
> [3] LCM-LoRA: A Universal Stable-Diffusion Acceleration Module.

---

> ### Comment · Reviewer_JFWr · 2025-11-25
>
> Thank you for your detailed answers to my questions and for sharing your code. I think your added experiments for motivation really help to sell and explain the method. I trust you will include them in your work, so I am raising my score.
>
> I still have some questions about the general research direction of learning a solver.
>
> > LCM-LoRA uses at least 60 million parameters [3, Table 1], whereas we train fewer than 220 parameters.
>
> What's the significance of this difference? Please correct me if I'm wrong here, but I don't think you are actually getting any kind of cost reduction here (on a per-token basis). While large compared to 220, 60 million parameters is miniscule in absolute terms (on the order of 1 GB to hold the params, gradients, and optimizer states). And as I mentioned, your method actually appears to require more compute and memory per training token because of having to run and store the activations of the entire solve.
>
> > Our training run requires 1-10 hours on an A100, whereas the fastest training run for LCM-LoRA requires approximately 32 hours on an A100.
>
> Although I don't have direct comparison to LCM-LoRA on hand, the comparison in Table 7a makes it seem like distillation-based methods substantially outperform training the solver. If time was equated (e.g. by taking fewer training steps during distillation), how do you think they would compare?

---

> ### Author Response · Authors · 2025-12-03
> **Official Comment by Authors - Part 1/2**
>
> Since reviewers JFWr and kJGM raised closely related questions, we address them together with one consolidated response that is repeated for each reviewer.
>
> We thank the reviewer for the detailed feedback. We would like to emphasize a fundamental distinction between GAS and distillation-based approaches. While distillation creates a new model by changing the weights of the original one, GAS trains only the solver, leaving the weights of the diffusion model unchanged. This key property allows GAS to fully preserve the generative characteristics of the original model.
>
> ## Distillation-based method uses at least 60 million parameters, whereas GAS uses fewer than 220 parameters. What's the significance of this difference?
>
> Research on the properties of generative models is presented in [1], which shows that Consistency Models may lag behind classical diffusion models in terms of subjective quality [1, Table 1] and diversity [1, Table 13]. Our experiments confirm this: GAS achieves higher diversity than distillation methods. Although the problem formulations differ, to directly address the reviewer's comment, we compared GS and GAS with distillation-based methods given equal training time and observed that our method demonstrates superior quality. We will add new comparison results with Consistency Models in Table 7a.
>
> To evaluate diversity, we compare the coverage [2] and FID of our method GS and GAS against the official checkpoint of CD [5] on CIFAR-10. We assess generation quality against a teacher solver [Appendix D.2], which achieves FID=2.03 and coverage=0.971. Throughout, when working with CD, we used the official implementation [3] and ternary search for NFE=4 [5, Section 3]. Coverage was measured between 10,000 generated images and the CIFAR-10 test set.
>
> The results of the comparison of FID and coverage are presented in Table 4 and show that GS and GAS have higher coverage than CD. In addition, CD demonstrates low coverage compared to the teacher solver, which indicates a deterioration in the generative properties of the diffusion model. Although 60M parameters are indeed relatively small, changing them through distillation can change the behavior of the model, which, as we have shown, harms diversity. Our method, without changing these parameters, provides a more flexible and property-preserving acceleration method.
>
> Table 4. Comparison of generative properties of models, CIFAR-10.
>
> |                      | CD, NFE=1 | CD, NFE=4 | GS, NFE=4 | GAS, NFE=4 | Our teacher |
> |----------------------|-----------|-----------|-----------|------------|-------------|
> | Coverage             | 0.942     | 0.938     |  0.963    |  0.961     |  0.971      |
> | FID                  | 3.56      | 2.99      |  4.41     |  4.05      |  2.03       |

---

> ### Author Response · Authors · 2025-12-03
> **Official Comment by Authors - Part 2/2**
>
> ## If time was equated (e.g. by taking fewer training steps during distillation), how do you think they would compare?
>
> Table 4 above compares fully-trained models. To directly address the reviewer's question about training cost, we additionally conducted experiments with matched training time, shown in Table 5 below. GS and GAS demonstrate higher generation quality for FID and coverage with equal training time. We trained CD from scratch using the official implementation [3] for 10,000 iterations, which took about 4 hours on 2 A100 GPUs. It took 19 GB of video memory and 4 hours of training on a single A100 to train GAS at NFE=4. Table 5 demonstrates that with similar training time, GS and GAS have better generation quality than CD at NFE=1 and NFE=4. In addition, GAS does not impose an overhead in time on inference compared to solvers; we show this in Appendix C.2.
>
> Table 5. Comparison of CD with GS, GAS with similar training time, CIFAR-10.
>
> |              | CD, NFE=1   | CD, NFE=4   | GS, NFE=4   | GAS, NFE=4   |
> |--------------|-------------|-------------|-------------|--------------|
> | Coverage     |  0.613      |  0.891      |  0.963      |  0.961       |
> | FID          |  55.24      |  8.64       |  4.41       |  4.05        |
> | Training time| 8h          | 4h          | 1h          | 3h           |
>
> These matched-time results directly address the reviewer's question: when training budgets are equalized, GS and GAS substantially outperform CD in both FID and coverage. For example, GAS achieves FID of 4.05 in 3 hours at NFE=4, while CD requires 4 hours to reach FID of 8.64, indicating that our solver-training approach is more sample-efficient during training while better preserving generative properties.
>
> Overall, we address the reviewer's concerns in two complementary ways. Conceptually, GAS trains only the solver while keeping the diffusion model’s weights fixed, whereas distillation modifies tens of millions of model parameters and can degrade generative behavior. Empirically, Tables 4 and 5 show that GS and GAS maintain higher coverage than CD at convergence and, under similar training time, achieve substantially better FID and coverage. Taken together, these results support our claim that trained solvers preserve the original model’s generative properties while offering superior efficiency under comparable compute budgets.
>
> ## General research direction of learning a solver.
>
> In future work, we plan to explore solvers that are conditioned on a class, prompt, or $x_t$. This direction can significantly expand the capabilities of trained solvers and open up new ways to accelerate generation while maintaining the generative properties of the original model. A promising application of trained solvers is their joint use with distillation-based methods. A striking example is Multistep Consistency Models [4], which enable integration with solvers [4, Section 3.1]. Further research into trained solvers will allow deeper exploration of the potential of such hybrid approaches, increasing their effectiveness and flexibility.
>
>
> ### References
>
> [1] Exposing flaws of generative model evaluation metrics
> and their unfair treatment of diffusion models
>
> [2] Reliable Fidelity and Diversity Metrics for Generative Models
>
> [3] https://github.com/openai/consistency_models_cifar10
>
> [4] Multistep Consistency Models
>
> [5] Consistency Models

---

### Public Comment · ~Shuai_Wang19 · 2025-11-28

Dear Authors, GAS optimizes both sampling parameters and distillation parameters, accelerating the inference of diffusion models. Our work, NeuralSolver[1], has been accepted by ICML 2025. In this paper, we have already explored search solver coefficients and sampling steps. However, your paper lacks a discussion of our work. Additionally, we have provided experiments on the joint training of distillation parameters and solver parameters in the Appendix.F (Table.6 and Table.7). Therefore, we believe it would be appropriate to compare and discuss our work.

[1] Wang, Shuai, et al. "Differentiable Solver Search for Fast Diffusion Sampling." arXiv preprint arXiv:2505.21114 (2025).

---

> ### Author Response · Authors · 2025-12-04
>
> Thank you for bringing NeuralSolver to our attention and congratulations on your acceptance to ICML 2025. We agree that both works share the core motivation of accelerating diffusion inference by optimizing solver parameters and timesteps via differentiable search through the sampling trajectory. We will definitely cite your work and include a detailed discussion and comparison in the revised version of our paper.
>
> While both methods optimize the sampling schedule and coefficients with a frozen diffusion model, we believe there are significant differences in formulation and objective that make them complementary:
>
> ## **Parameterization & Initialization:**
>
> - NeuralSolver employs a constrained linear multistep formulation where the diagonal coefficient is determined by $c_{i}^{i} = 1 - \sum c_{j}^{i}$ to ensure consistency. It initializes parameters to represent an Euler solver (zeroed off-diagonals) with uniform timesteps and learns the coefficients from scratch to minimize trajectory error.
>
> - GAS introduces a Generalized Solver parameterization defined as additive corrections on top of a strong, high-order base solver (DPM-Solver++). We initialize our search from the optimal DPM-Solver++ coefficients and schedule, rather than starting from Euler/Uniform, allowing us to retain the benefits of the analytical baseline while adapting to specific data distributions.
>
> ## **Training Objective and Fidelity:**
>
> - NeuralSolver focuses on minimizing discretization error by regressing to a high-NFE teacher trajectory using MSE and Huber loss. This ensures high faithfulness to the ODE path but does not explicitly optimize for perceptual quality.
>
> - GAS targets the few-step regime (4-8 NFE) by combining distillation with Adversarial (GAN) and Perceptual (LPIPS) losses. This allows our method to prioritize perceptual fidelity and texture details even when the exact ODE trajectory cannot be perfectly preserved due to the low step count.
>
> ## **Scope:**
>
> - Your work demonstrates impressive results on ImageNet (DiT/Rectified Flow) around 10 NFEs. Our work focuses on pushing the limit of standard latent diffusion models (such as SD1.5) and pixel-space models down to 4-8 NFEs, where the adversarial component becomes critical for maintaining sharp details.
>
> To ensure a fair and accurate comparison on our benchmarks (Stable Diffusion / ImageNet 256x256), could you please share the official implementation of NeuralSolver? Since our code is already public, having access to your codebase would allow us to reproduce your exact setup (including the specific initialization and regularization hyperparameters) and report the most reliable metrics in our revision.

---

### Author Response · Authors · 2025-12-03
**Summary for Area Chair(s): Reviewers' Positive Feedback and Addressed Concerns (1/2)**

Dear Reviewers and Area Chair,

We sincerely thank the reviewers for their thoughtful engagement and constructive feedback throughout the discussion period. Your suggestions will help us make the camera-ready version clearer and further strengthen the presentation of our method’s effectiveness.

The reviewers highlighted several key strengths of our contribution:

- **A logical and efficient formulation**, focusing on training the solver rather than the full generative model;

- **A flexible solver parameterization** that yields notable performance gains. We emphasize that this is one of our key contributions;

- A natural and intuitive **integration of adversarial loss**, which substantially enhances distribution alignment and boosts generation quality — especially in low-NFE regimes;

- **Comprehensive ablations** that clarify the role of each component, along with strong cross-dataset generalization.

During the discussion, we addressed concerns regarding the practical advantages of our approach compared to other distillation methods, conducted controlled comparisons with S4S [1] under fully matched settings and clarified the motivation behind our design choices with additional ablations. We are particularly encouraged that reviewer JFWr is ready to raise his score after our comprehensive response. Below we summarize how we addressed each major concern and the experimental evidence supporting our method's effectiveness.

&nbsp;

### __Practical advantages over other distillation methods__

Reviewers kJGM and JFWr raised concerns about whether our trained-solver approach offers practical advantages over other distillation methods, such as Consistency Distillation (CD) [2]. We argue that solver distillation better preserves the teacher’s generative characteristics, offers fundamental methodological advantages, and performs better under equal training-time constraints.

- **Fundamental methodological advantage**

Because GS/GAS keep the diffusion model’s weights fixed and learn a compact solver with an exceptionally small number of trainable parameters, they preserve the teacher’s generative characteristics—including diversity, coverage, and fine-grained details. In contrast, distillation methods train a new student network and modify the generator’s weights, which can compromise these properties.

- **Key experimental findings**

Under equalized training budgets on the CIFAR-10 dataset, GAS reaches **FID of 4.05 in 3 hours** at NFE = 4, whereas CD achieves **8.64 in 4 hours** at the same NFE—demonstrating substantially better quality in comparable or less time. Moreover, using the Reliable Fidelity and Diversity Metrics [3], GAS obtained higher Coverage than CD (0.961 vs 0.942), indicating better sample diversity without mode collapse.

&nbsp;

### __Detailed S4S comparison under the same settings__

Reviewer bLXP raised a concern that the comparison with S4S lacked sufficient experimental controls, making it difficult to isolate GAS’s true advantages. Because S4S does not provide an official implementation and omits several hyperparameter details, we reimplemented the method and conducted extensive controlled experiments using our fixed hyperparameter configuration.

- **Fully matched experimental setup**

We compared GS with S4S-style parameterizations (LMS and LMS+PC) under strictly identical settings: the same teacher trajectories, training hyperparameters, and initialization. Across all evaluated NFEs on FFHQ and ImageNe, GS consistently outperformed both S4S parameterizations (LMS and LMS+PC) under these rigorously controlled conditions.

- **Analysis of the alternating training strategy.**

Using our implementation of the S4S alternating-training procedure, we found that alternation significantly improves S4S LMS+PC, but does not improve GS or standard S4S LMS. This indicates that GS’s parameterization removes the reliance on the complex and often unstable alternating-training strategy, delivering both better performance and a simpler, more stable training process.

---

> ### Author Response · Authors · 2025-12-03
> **Summary for Area Chair(s): Reviewers' Positive Feedback and Addressed Concerns (2/2)**
>
> ### __Motivation and Design Choices__
>
> Reviewers JFWr and bLXP asked for clearer motivation behind GS’s design choices and additional ablations validating each component. We addressed each point as follows:
>
> - **Theory-based initialisation as a strong inductive bias**
>
> Instead of learning solver coefficients from scratch, GS learns additive corrections on top of theory-derived coefficients—specifically those from DPM-Solver++(3M)—which we treat as theoretical guidance. Since these coefficients are time-dependent, and the timesteps themselves are trainable, both the base coefficients and their learned corrections are updated whenever the timesteps change. Such technique significantly stabilizes training and improves convergence. In our ablations, it yields substantially lower FID across NFEs on FFHQ compared to training without such a theoretical foundation.
>
> - **Use of the linear multistep (LMS) signature**
>
> We adopt the LMS structure because it generalizes a broad class of classical ODE solvers. This provides a principled, theory-aligned foundation for GS rather than relying on an arbitrary architectural choice.
>
> - **Additive parameterization**
>
> The form $c = c_{\text{theory}} + \Delta c$ of the the solver coefficients allows GS to be interpreted as learning residual corrections on top of a stable theoretical solver—conceptually similar to residual connections or LoRA. Ablations show that additive parameterization achieves slightly better performance than multiplicative alternatives.
>
> - **Relativistic GAN loss**
>
> We demonstrate that GAS performs well with both traditional and relativistic adversarial losses. We prefer the letter because it aligns with current state-of-the-art GAN training practices and yields superior generation quality in our experiments.
>
> &nbsp;
>
> ### __Additional Experiments__
>
> - **Absence of mode collapse**
>
> Using the Reliable Fidelity and Diversity Metrics evaluation, we verified that incorporating of the adversarial loss does not introduce any mode-collapse issues. Coverage metrics remain high, indicating that the adversarial objective improves generation quality without reducing sample diversity.
>
> - **No alternating strategy required**
>
> Our ablation demonstrates that, while alternating training improves S4S’s LMS+PC parameterization, it actually worsens GS performance. This highlights that GS’s parameterization inherently provides stable optimization and eliminates the need for any alternating strategy.
>
> &nbsp;
>
> &nbsp;
>
> We believe that these comprehensive responses—supported by rigorous experimental evidence—address all major reviewer concerns and clearly demonstrate both the theoretical soundness and practical advantages of our approach. Thank you for your consideration.
>
> &nbsp;
>
> ### References
>
> [1] Eric Frankel, et al. "S4S: Solving for a Diffusion Model Solver."
>
> [2] Yang Song, et al. "Consistency Models."
>
> [3] Muhammad Ferjad Naeem, et al. "Reliable Fidelity and Diversity Metrics for Generative Models."

---

### Meta-Review · Area_Chair_UK3W · 2025-12-13

**Summary:**

Summary of reviewers’ concerns:
* Lacking motivation for the GS solver and ablations for the design choices, e.g. additive coefficients (reviewers JFWr and bLXP)
* Initialization of DPM-Solver++ solver,  whether it is theoretically-justified to start from LMS-type solver (reviewers bLXP and kJGM).
* Completeness of comparisons to baselines, particularly vs S4S (reviewer JFWr, kJGM). The comparisons need to be in exact settings and might require  retraining of S4S using the authors’ exact teacher and initialization.
* Practical advantages over distillation/PEFT methods (reviewers JFWr and kJGM)
* Possible mode collapse from adversarial loss (reviewer bLXP)

**Reviewer Concerns:**

Addressed concerns:
* Motivation / GS design and ablations (reviewers JFWr, bLXP). Authors added motivation  and provided extensive ablations, including showing theoretical coefficients significantly improve quality
* S4S comparison (bLXP). Addressed by authors’ matched reimplementation and alternating-training ablation; GS still outperforms S4S in their matched runs.
* GAN loss choice and mode collapse (JFWr, bLXP, kJGM). Validated that no mode collapse occurring due to the GAN loss.
* DPMSolver++ initialization (kJGM). The authors explained the theory behind initializing DPM-Solver++ solver, motivating that it is similar to Adams-Bashforth-style multistep update.



Remaining concerns:
* Cost in comparison to PEFT/distillation: Reviewers JFWr and kJGM questioned the authors' claim that having fewer parameters (220 vs 60M) inherently makes the method more computationally efficient than PEFT/LoRA.  Authors provided matched-time comparisons vs CD, but the reviewer kJGM remained unconvinced in the thread.

**Reviewer Scores:**

The authors gave satisfactory answers to almost all of reviewer's concerns. If the reviewers could participate in the discussion, the scores might have changed as follows:

* Reviewer JFWr, score 2 – increase to 6
* Reviewer bLXP, score 4 – increase to 6
* Reviewer kJGM, score 6 – unchanged.

---

### Decision · Program_Chairs · 2026-01-26

Accept (Poster)